https://doi.org/10.1038/s41467-019-09954-9　　**OPEN**

# Impaired cortico-striatal excitatory transmission triggers epilepsy

Hiroyuki Miyamoto[1,2,3,11], Tetsuya Tatsukawa[1,11], Atsushi Shimohata[1], Tetsushi Yamagata[1], Toshimitsu Suzuki[1], Kenji Amano[1], Emi Mazaki[1], Matthieu Raveau[1], Ikuo Ogiwara[1,4], Atsuko Oba-Asaka[3,5], Takao K. Hensch[3], Shigeyoshi Itohara[5,6], Kenji Sakimura[7], Kenta Kobayashi[8,9], Kazuto Kobayashi[10] & Kazuhiro Yamakawa[1]

*STXBP1* and *SCN2A* gene mutations are observed in patients with epilepsies, although the circuit basis remains elusive. Here, we show that mice with haplodeficiency for these genes exhibit absence seizures with spike-and-wave discharges (SWDs) initiated by reduced cortical excitatory transmission into the striatum. Mice deficient for *Stxbp1* or *Scn2a* in cortico-striatal but not cortico-thalamic neurons reproduce SWDs. In *Stxbp1* haplodeficient mice, there is a reduction in excitatory transmission from the neocortex to striatal fast-spiking interneurons (FSIs). FSI activity transiently decreases at SWD onset, and pharmacological potentiation of AMPA receptors in the striatum but not in the thalamus suppresses SWDs. Furthermore, in wild-type mice, pharmacological inhibition of cortico-striatal FSI excitatory transmission triggers absence and convulsive seizures in a dose-dependent manner. These findings suggest that impaired cortico-striatal excitatory transmission is a plausible mechanism that triggers epilepsy in *Stxbp1* and *Scn2a* haplodeficient mice.

[1] Laboratory for Neurogenetics, RIKEN Center for Brain Science, Wako, Saitama 351-0198, Japan. [2] PRESTO, Japan Science and Technology Agency, Saitama 332-0012, Japan. [3] International Research Center for Neurointelligence (IRCN), The University of Tokyo Institutes for Advanced Study, Tokyo 113-0033, Japan. [4] Department of Physiology, Nippon Medical School, Tokyo 113-8602, Japan. [5] Laboratory for Behavioral Genetics, RIKEN Center for Brain Science, Wako, Saitama 351-0198, Japan. [6] FIRST, Japan Science and Technology Agency, Saitama 332-0012, Japan. [7] Department of Cellular Neurobiology, Brain Research Institute, Niigata University, Niigata 951-8585, Japan. [8] Section of Viral Vector Development, National Institute for Physiological Sciences, Okazaki 444-8585, Japan. [9] Graduate University for Advanced Studies (SOKENDAI), Hayama 240-0193, Japan. [10] Department of Molecular Genetics, Institute of Biomedical Sciences, Fukushima Medical University School of Medicine, Fukushima 960-1295, Japan. [11]These authors contributed equally: Hiroyuki Miyamoto, Tetsuya Tatsukawa. Correspondence and requests for materials should be addressed to K.Y. (email: kazuhiro.yamakawa@riken.jp)

Mutations in *STXBP1*, which encodes Munc18-1, a pre-synaptic protein essential for neurotransmitter release, and *SCN2A*, which encodes Nav1.2, a voltage-gated sodium channel alpha II subunit, are common in patients with a wide spectrum of neurological disorders, including epilepsy, intellectual disability, autism, and schizophrenia[1–7]. In particular, *STXBP1* and *SCN2A* mutations are common in patients with early-infantile epileptic encephalopathy (Ohtahara syndrome), West syndrome and Lennox-Gastaut syndrome[2,3,8–10], suggesting a potentially overlapping pathological mechanism.

*Stxbp1*-haplodeficient (*Stxbp1$^{+/-}$*) mice display emotional[11] and spatial[12] learning and memory deficits and enhanced aggression[11]. Additionally, *Scn2a*-haplodeficient (*Scn2a$^{+/-}$*) mice exhibit deficits in spatial learning and memory[13]. Therefore, these behavioral phenotypes are consistent with the cognate human syndrome. Moreover, *Stxbp1$^{+/-}$* mice[14] and *Scn2a$^{+/-}$* mice[15] show spike-and-wave discharges (SWDs) on electrocorticograms (ECoG).

Slow SWDs (1.5−2.5 Hz) are observed in patients with Ohtahara syndrome[16] and Lennox-Gastaut syndrome[17], which show generalized tonic, tonic-clonic, myoclonic, atypical absence and atonic seizures, while 3 Hz SWDs are typically observed in patients with absence epilepsy[18]. There are several rodent models of absence epilepsy in which thalamocortical circuits have been widely accepted as the primary generator of SWDs[19,20]. To date, debate continues concerning the critical contributions of cortico-thalamic or thalamo-cortical neurons in generating SWDs within thalamocortical circuits[19,21–23]. The somatosensory cortex (SSC) has been proposed as the site of initial appearance of SWDs in rat models of absence epilepsy[24,25]; however, the initial changes responsible for SWD generation and their associated mechanisms have not been fully elucidated.

In this study, we investigated the initial triggers for SWDs in *Stxbp1$^{+/-}$* and *Scn2a$^{+/-}$* mice and their associated circuit mechanisms. Contrary to the previous proposal of the basal ganglia as merely a modulator of SWDs primarily produced by thalamocortical circuits[26–28], we here show that impairments in cortico-striatal, rather than cortico-thalamic, pathways trigger SWDs in *Stxbp1$^{+/-}$* and *Scn2a$^{+/-}$* mice using multiple experimental approaches.

## Results

***Stxbp1$^{+/-}$* mice show absence seizures.** We generated multiple developmental and conditional knockout lines of both *Stxbp1$^{+/-}$* and *Scn2a$^{+/-}$* mice in order to model the diseases associated with these genes and identify the underlying circuit mechanism. As reported previously[14], we verified that adult *Stxbp1$^{+/-}$* mice[11] exhibited SWDs in SSC ECoG (Fig. 1a and Supplementary Fig. 1a–g) and occasional myoclonic or generalized convulsive seizures with vocalizations (Supplementary Video 1). Intraperitoneal injection of ethosuximide, a T-type Ca$^{2+}$ channel blocker, into *Stxbp1$^{+/-}$* mice eliminated SWDs in the SSC (Fig. 1b), suggesting that thalamic[29], cortical[30], or subthalamic T-type Ca$^{2+}$ channels are necessary for SWD generation. Similar to other rodent models of absence epilepsy[19,20] and *Scn2a$^{+/-}$* mice[15], *Stxbp1* knockout mice showed synchronous bilateral cortical SWDs during behavioral quiescence (Fig. 1a) and effective suppression of SWDs following ethosuximide administration (Fig. 1b), therefore they were regarded as experiencing absence seizures.

Multisite local field potential (LFP) recordings in *Stxbp1$^{+/-}$* mice detected SWDs in the medial prefrontal cortex (mPFC) and dorsal striatum (caudate putamen: CPu), concurrent with the SSC-SWDs (Fig. 1c and Supplementary Fig. 1b). Moreover, bipolar recordings in the SSC and CPu suggested that SWD generation occurred within local circuits (Supplementary Fig. 1f). Stable temporal phase relationships among SWDs in the SSC, mPFC, CPu, and ventroposterior thalamus (Thal) imply neuronal interactions among these regions (Supplementary Fig. 2a). SWD peaks in the mPFC, CPu and Thal were equally delayed compared to those observed in SSC ECoG recordings (Supplementary Fig. 2a, b), although the initial appearance of SWDs in the CPu slightly preceded those in the SSC (Fig. 1c and Supplementary Fig. 2c, d).

Kovacevic et al.[14] reported that *Stxbp1$^{+/-}$* mice show spontaneous twitches and jumps concurrent with SWDs displaying positive polyspikes. In the present study, we also observed involuntary twitches (5–8 times/6 h) and jumps (3–5 times/6 h) during sleep (Supplementary Video 2). However, SWDs, which had negative peaks, did not concur with the twitches or jumps, but rather with behavioral quiescence [91.3%; 73 out of 80 SWD episodes, $N = 4$ mice] (Fig. 1a). The twitches or jumps occurred with positive deflections in the ECoG recordings (Supplementary Fig. 3). We observed SWDs not only during quiet waking, but also during non-rapid eye movement (REM) and REM sleep in *Stxbp1$^{+/-}$* mice (Supplementary Fig. 4).

**Striatum as a critical node for epilepsy.** We investigated the neural circuits required to generate SWDs in the mutant mice. Local injection of muscimol, a GABA$_A$ receptor agonist, into the SSC, CPu, or Thal but not into the mPFC or hippocampal CA1 region suppressed SWDs in SSC ECoG recordings in *Stxbp1$^{+/-}$* mice (Fig. 2a, left; Supplementary fig. 5a). In *Stxbp1$^{+/-}$* mice receiving CPu injection, SWDs were well suppressed not only in the SSC but also in the mPFC and CPu (Fig. 2a, right) where strong SWDs were observed before injection (Fig. 1c). These results demonstrate that neural activities in the SSC, CPu, and Thal are required for the generation or maintenance of SWDs in *Stxbp1$^{+/-}$* mice. Although the SSC and thalamus have been well recognized as critical nodes for SWD generation, these results indicate that the CPu is also crucial; thereafter it became a focus of our subsequent experiments. In contrast to muscimol, micro-injection of bicuculline, a GABA$_A$ receptor antagonist, into the CPu of *Stxbp1$^{+/-}$* mice induced myoclonic and subsequent generalized convulsive seizures (Supplementary Fig. 5b). These data suggest that the CPu is involved in the generation of both absence (non-convulsive) and convulsive seizures. Indeed, short duration monophasic electrical stimulation (single pulse) applied to the CPu of one hemisphere of the brain in adult *Stxbp1$^{+/-}$* mice using a depth electrode, triggered generalized SWDs in the SSC, mPFC, CPu, and Thal (Fig. 2b left), and we observed similarities between spontaneous (Supplementary Fig. 2a) and evoked SWDs (Supplementary Fig. 6) in their waveforms and phase relationships. Additionally, brain regions with a higher occurrence of spontaneous SWDs (Fig. 1c) also showed a higher occurrence of evoked SWDs (Fig. 2b, right). These results suggest that brief and local activation of the CPu is sufficient to generate SWDs in *Stxbp1$^{+/-}$* mice.

**Impaired cortico-striatal excitatory inputs causes epilepsy.** To evaluate the comparative impact of *Stxbp1* deletion in excitatory and inhibitory neurons on absence seizures, we generated *Stxbp1* conditional knockout mice using *Emx1*-Cre recombinase (*Emx1*-Cre)[11,31,32]-mediated dorsal-telencephalic (i.e., the cerebral cortex, hippocampus, and amygdala, but excluding the striatum, globus pallidus, and thalamus) excitatory neuron-specific deletion or *Vgat*-Cre[11,32]-mediated global inhibitory neuron-specific deletion. Notably, *Stxbp1$^{flox/+}$*/*Emx1*-Cre (*Stxbp1$^{fl/+}$/Emx*) mice reproduced SWDs with negative peaks during behavioral quiescence (Fig. 3a) but did not show twitches or jumps. In contrast,

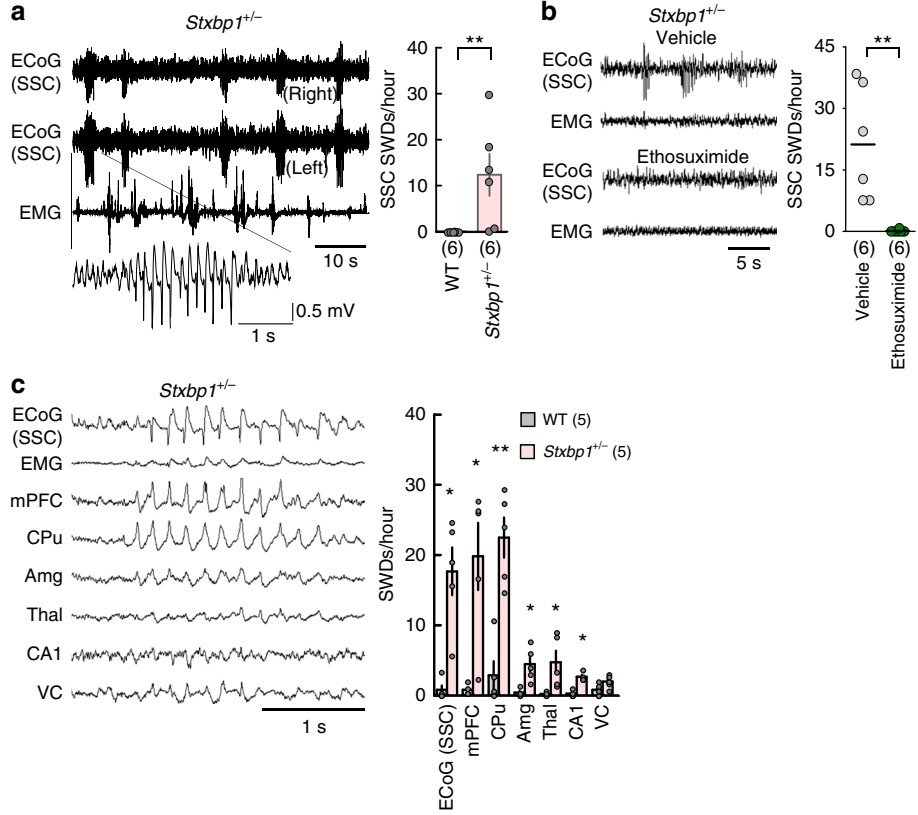

**Fig. 1** Ethosuximide-sensitive SWDs predominantly appear in the SSC, mPFC and CPu of $Stxbp1^{+/-}$ mice. **a** SSC SWDs in $Stxbp1^{+/-}$ mice. (Left) Bilateral ECoG and electromyogram (EMG) recordings in an $Stxbp1^{+/-}$ mouse. (Right) Number of SWDs (24 h recordings). WT ($N = 6$) and $Stxbp1^{+/-}$ mice ($N = 6$) (Mann–Whitney $U$ test; **$P = 0.0047$). **b** Ethosuximide efficiently suppressed SWDs. (Left) Representative recordings. (Right) Number of SWDs (2.5 h recordings). $Stxbp1^{+/-}$ mice ($N = 6$) (Mann–Whitney $U$ test; vehicle vs. ethosuximide, **$P = 0.005$). **c**, SWDs predominantly appeared in the SSC, mPFC, and CPu of $Stxbp1^{+/-}$ mice. (Left) Simultaneously recorded SWDs in multiple brain regions. (Right) Number of SWDs during the light period (3-hours recording). WT ($N = 5$) and $Stxbp1^{+/-}$ ($N = 5$) mice. Statistical comparisons between WT and $Stxbp1^{+/-}$ mice were made in each brain region (Mann–Whitney $U$ test, WT vs. $Stxbp1^{+/-}$, SSC: *$P = 0.0117$; mPFC: *$P = 0.0112$; CPu: **$P = 0.0079$; Amg: *$P = 0.0117$; Thal: *$P = 0.0112$; CA1: *$P = 0.0112$). Data represent mean ± SEM. *$P < 0.05$, **$P < 0.01$. Mouse numbers in parentheses. mPFC medial prefrontal cortex, CPu caudate putamen, Amg basolateral amygdala, Thal ventroposterior thalamus (a part of somatosensory system), CA1 hippocampus CA1 region, VC visual cortex

we observed that $Stxbp1^{flox/+}/Vgat$-Cre ($Stxbp1^{fl/+}/Vgat$) mice showed twitches (~ 4 times over 6 h) and jumps (3–5 times over 6 h) (Supplementary Video 3) coinciding with ECoG-positive deflections (Supplementary Fig. 3b) but not with SWDs (Fig. 3a, right) or any other epileptic phenotypes. Although mice with a conditional haplo-deletion of $Stxbp1$ in inhibitory neurons using a $Gad2$-Cre driver ($Gad2$-$Stxbp1^{cre/+}$) have been reported to show severe epileptic phenotypes and a low survival rate[14], we observed that $Stxbp1^{flox/+}/Vgat$-Cre ($Stxbp1^{fl/+}/Vgat$) mice showed a normal survival rate, normal growth and locomotor ability[11]. Our results clearly indicate that $Stxbp1$-haploinsufficiency in dorsal-telencephalic excitatory neurons is responsible for SWDs during behavioral quiescence, while the same condition in GABAergic neurons is responsible for the twitches/jumps.

Additionally, we found that $Scn2a^{+/-}$ and $Scn2a^{fl/+}/Emx$ mice but not $Scn2a^{fl/+}/Vgat$ mice showed SWDs during behavioral quiescence, although these were milder than those in $Stxbp1^{+/-}$ mice[15], suggesting an overlapping pathological circuit for absence seizures in $Stxbp1^{+/-}$ and $Scn2a^{+/-}$ mice. Similarly to $Stxbp1^{+/-}$ mice, muscimol injections into the CPu or Thal suppressed SWDs in $Scn2a^{fl/+}/Emx$ mice (Supplementary Fig. 7).

Microdialysis analysis revealed that basal glutamate release, but not GABA release (normalized against high $K^+$-evoked maximal release), was significantly lower in the CPu of behaving $Stxbp1^{+/-}$ mice than in wild-type (WT) mice (Fig. 3b, c). Ampakine (CX516) potentiates postsynaptic AMPA receptors in the presence of glutamate[33]. Intraperitoneal administration (Fig. 3d) or local injection of ampakine into the CPu, but not into the thalamus, significantly reduced SWDs (Fig. 3e). These results indicate that impaired cortico-striatal excitatory synaptic transmission is responsible for the generation of SWDs and epileptic seizures in $Stxbp1^{+/-}$ mice.

To confirm the specific contribution of cortico-striatal synaptic transmission to SWD generation, we engineered mice with a conditional deletion of $Stxbp1$ or $Scn2a$ using a $Trpc4$-Cre driver mouse line[34] expressing Cre-recombinase in cortical layer 5 pyramidal neurons, which are a major source of excitatory inputs to the striatum (Fig. 4a left; Supplementary Fig. 8a, b), or a $Ntsr1$-Cre mouse[23] expressing Cre-recombinase in cortical layer 6 neurons, which project to the thalamus (Fig. 4a right). Notably, $Stxbp1^{flox/flox}/Trpc4$-Cre ($Stxbp1^{fl/fl}/Trpc$) mice, but not $Stxbp1^{flox/flox}/Ntsr1$-Cre ($Stxbp1^{fl/fl}/Ntsr$) mice, displayed SWDs (Fig. 4b). Similarly, $Scn2a^{fl/fl}/Trpc$ mice, but not $Scn2a^{fl/fl}/Ntsr$ mice, displayed SWDs (Fig. 4c, Supplementary Fig. 8c). By contrast, $Ntsr1$-Cre dependent deletion of $Cacna1a$ gene encoding a P/Q-type voltage-gated calcium channel α subunit was reported to produce SWDs in mice[23]. These data suggest that different gene deletions exhibit different effects on downstream circuitry for SWDs (i.e. cortico-thalamic vs. cortico-striatal).

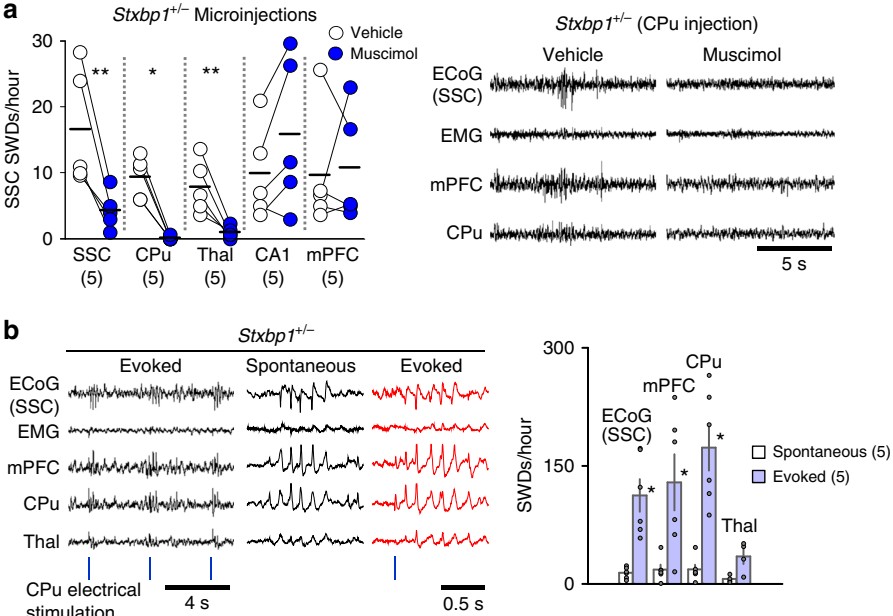

**Fig. 2** Inactivation of CPu, SSC and thalamus (Thal) suppresses SWDs and activation of CPu induces SWDs in *Stxbp1*$^{+/-}$ mice. **a** Muscimol injections in *Stxbp1*$^{+/-}$ mice. (Left) Injections into SSC, CPu, Thal but not in CA1 and mPFC suppressed SWDs in SSC ECoG recordings (3 h recording after injection). SSC injection ($N = 5$), CPu injection ($N = 5$), Thal injection ($N = 5$), CA1 injection ($N = 5$), mPFC injection ($N = 5$). Mann–Whitney $U$ test, vehicle vs. muscimol, SSC: **$P = 0.0079$; CPu: *$P = 0.0119$; Thal: **$P = 0.0079$; CA1, $P = 0.5476$, mPFC $P = 0.9166$. Numbers of mice are shown in parentheses. (Right) Representative recordings in the mice with CPu injection. SWDs were suppressed in SSC, mPFC and CPu recordings. **b** Single brief electrical stimulation of the CPu evoked SWDs in *Stxbp1*$^{+/-}$ mice. (Left) Representative recordings. (Right) Regional occurrence of spontaneous and evoked SWDs (6 experiments from 5 mice, 1 h recording). Mann–Whitney $U$ test, spontaneous SWDs vs. evoked SWDs, SSC: **$P = 0.0022$; mPFC: *$P = 0.0152$; CPu: **$P = 0.0022$; Thal: $P = 0.0571$. Mean ± SEM. *$P < 0.05$, **$P < 0.01$

We then used the NeuRet system[35,36] to generate mice with an *Stxbp1* deletion restricted to cortico-striatal projection neurons (Supplementary Fig. 9). A retrograde lentivirus containing a flippase (FLP) gene was injected into the CPu of adult WT mice (>2 months) and was taken up by axons terminating in the CPu (Fig. 4d). Subsequent injection of an adeno-associated virus (AAV) containing a FLP-dependent double-inverted-orientation Cre (fDIO-Cre) gene into the SSC allowed SSC neurons projecting to the CPu to express the Cre gene. The floxed *Stxbp1* genes in cortico-striatal projecting neurons were then excised (Supplementary Fig. 9). As expected, the lentivirus and AAV-injected *Stxbp1*$^{fl/fl}$ mice exhibited SWDs, whereas the virus-injected *Stxbp1*$^{fl/+}$ or WT mice did not (Fig. 4e). These data indicate that *Stxbp1* deletion in cortico-striatal projection neurons, even in the adult stage, is sufficient to cause SWDs.

**Striatal fast-spiking interneurons control epilepsy.** In the CPu, both striatal medium spiny neurons (MSNs) and fast-spiking interneurons (FSIs) receive excitatory inputs from neocortical pyramidal neurons, and the cortex exerts potent feed-forward inhibition on MSNs via FSIs[37]. We measured synaptic drive to MSNs and striatal FSIs in whole-cell recordings of cortico-striatal brain slices from *Stxbp1*$^{+/-}$ mice. After electrophysiological identification of MSNs and FSIs (Supplementary Fig. 10a, b), excitatory postsynaptic currents (EPSCs) were evoked by electrical stimulation in the SSC neocortical layer 5/6 (Fig. 5a). FSIs can be reliably identified by their characteristic fast-spiking pattern (>200 Hz) with short spike width (<1 ms), clear after-hyperpolarization, and minimal firing adaptations[38] (Supplementary Fig. 10a). Notably, we found a significantly faster rundown of EPSCs in the putative FSIs but not in the MSNs of *Stxbp1*$^{+/-}$ mice, at 10- to 40-Hz stimulation as compared with

that observed in WT mice (Fig. 5b–d, Supplementary Fig. 11a). The absolute amplitude of the initial EPSCs, the first EPSCs evoked by current injection, in WT and *Stxbp1*$^{+/-}$ mice did not differ significantly, and the amplitudes of asynchronous miniature EPSCs in FSIs and MSNs before and after stimulation also did not change, confirming the presence of unaltered postsynaptic sensitivity (Supplementary Fig. 11b, c). These results indicate that excitatory presynaptic transmission was predominantly impaired in neocortical–striatal FSI connections. The decrease in EPSCs in the FSIs of *Scn2a*$^{+/-}$ mice was not significant (Supplementary Fig. 11d, e), likely due to the direct involvement of sodium channels in action-potential generation rather than in synaptic transmission itself, which might reflect the lower SWD occurrence in *Scn2a*$^{+/-}$ mice[15] relative to that in *Stxbp1*$^{+/-}$ mice. Additionally, the estimated size of the readily releasable pool (RRP) of synaptic vesicles, i.e., the assembly of synaptic vesicles filled with neurotransmitters that are docked, primed, and ready for exocytosis, did not differ between WT and *Stxbp1*$^{+/-}$ mice (Supplementary Fig. 12a–c). We found that the size of the RRPs in FSIs was significantly larger than those in MSNs in both WT and *Stxbp1*$^{+/-}$ mice. Munc18-1 haploinsufficiency might limit the replenishment of synaptic vesicles from reserve or recycling pools to the RRP[39] or affect a downstream step of vesicle priming in the RRP[40]; however the reason for the dominant impairment in neocortical-striatal FSI connections remains unknown.

To mimic impaired cortical excitatory input to striatal FSIs and reduce their activity in vivo, we injected 1-naphthyl acetyl spermine (NASPM), a selective blocker of calcium-permeable AMPA receptors and abundantly expressed in striatal FSIs but not in MSNs[41–43], into the CPu of WT mice (Fig. 6a). At a low dose (5 mM, 0.2 μl, bilateral), SWDs appeared in the CPu and mPFC (Fig. 6b, c) and those with longer durations occasionally appeared in SSC-ECoG recordings (Supplementary Fig. 13a), with

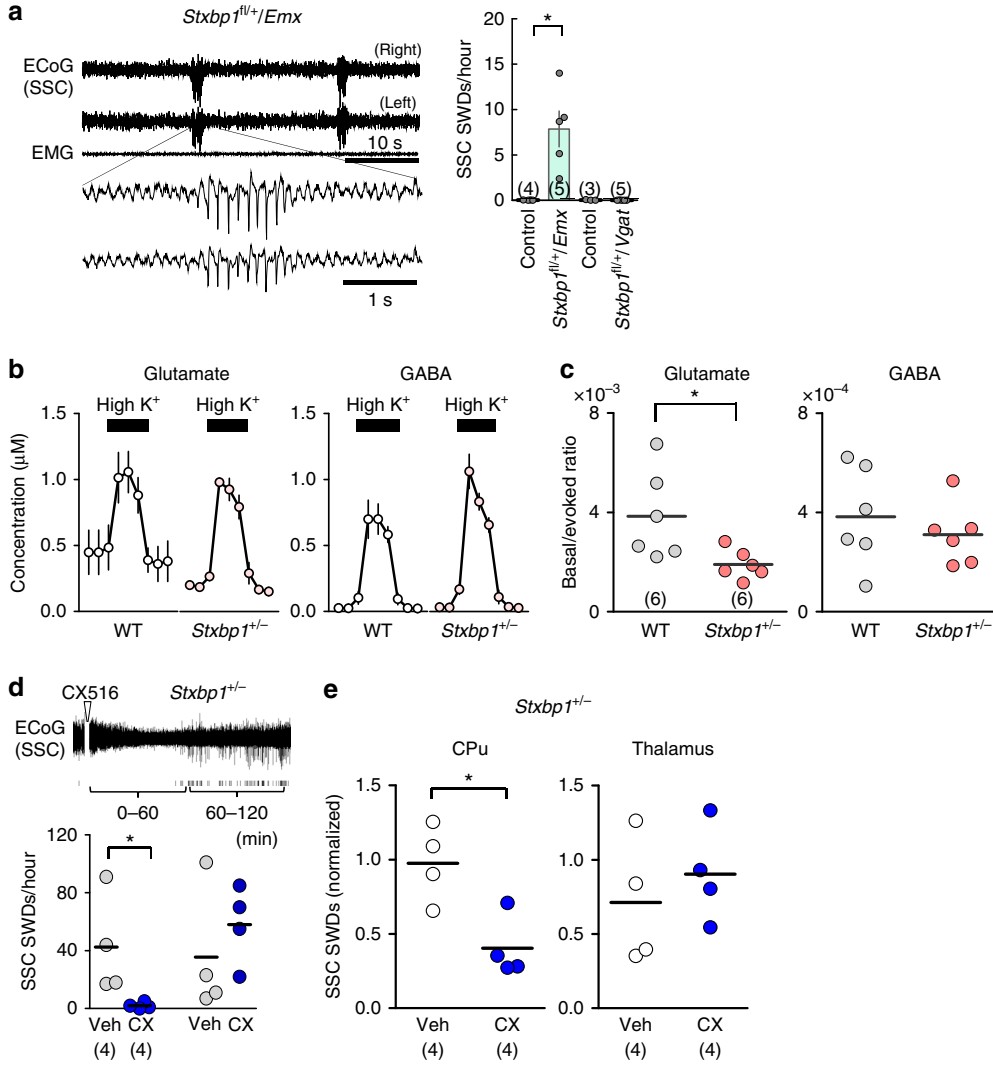

**Fig. 3** Impaired cortico-striatal excitatory transmission underlies SWDs in *Stxbp1*+/− mice. **a** SWDs appeared in *Stxbp1*fl/+/*Emx*, but not in *Stxbp1*fl/+/*Vgat* mice. (Left) Representative SWDs in an *Stxbp1*fl/+/*Emx* mouse. (Right) Number of SSC SWDs (24 h recording). *Stxbp1*fl/+/*Emx* (N = 5), littermate controls (1 WT, 2 *Stxbp1*fl/+, 1 *Stxbp1*+/+/*Emx*, N = 4); *Stxbp1*fl/+/*Vgat* (N = 5), littermate controls (1 WT, 2 *Stxbp1*fl/+, N = 3). Mann–Whitney *U* test, control vs. *Stxbp1*fl/+/*Emx*: *P = 0.0286; control vs. *Stxbp1*fl/+/*Vgat*: P = 0.8255. **b** Glutamate and GABA release in the CPu of freely behaving WT (N = 6) and *Stxbp1*+/− (N = 6) mice with high K+ stimulation. **c** Basal glutamate release (normalized by high K+-evoked release), but not GABA release, was significantly lower in *Stxbp1*+/− mice. Mann–Whitney *U* test, WT vs. *Stxbp1*+/−, glutamate: *P = 0.026; GABA P = 0.5887. **d** Potentiation of AMPA receptors in the CPu suppressed SWDs in *Stxbp1*+/− mice. (Top) Representative SSC-ECoG after intraperitoneal injection of the ampakine, CX516. Tick marks represent SWDs. (Bottom) SWDs were suppressed in the first, but not in the second, 60 min following CX516 intraperitoneal injection. *Stxbp1*+/− mice (N = 4), vehicle vs. CX516, significant in the first but not the second 60 min period (Mann–Whitney *U* test, vehicle vs. CX516, 0–60 min: *P = 0.0268; 60–120 min: P = 0.4857). **e** CX516 injection into the CPu, but not the thalamus, suppressed SWDs (3000 s recording after injection, normalized, see Methods), Mann–Whitney *U* test, vehicle vs. CX516, CPu: *P = 0.0317; Thal: P = 0.4357). Calculation using unnormalized data in CPu had an tendency of decrease in SWDs but did not reach to a statistical significance. Mean ± SEM (**a**). *P < 0.05. Numbers of mice are shown in parentheses

this finding consistent with that in a previous study[43]. Similar to previous observations of dyskinetic movements in mice with selective inhibition of striatal FSIs[42,43], NASPM sequentially induced behavioral quiescence, myoclonic, dyskinesia-like tonic, clonic, and generalized tonic-clonic seizures within 1 h of administration and in a dose-dependent manner (Fig. 6b–d, Supplementary Video 4). Epileptic activities in the ECoG and LFP occurred concurrently with generalized seizures (Fig. 6b, right). Long SWDs in ECoG recordings and epileptic convulsive seizures were also observed when NASPM was injected into the reticular thalamic nucleus of WT mice, where abundant calcium-permeable AMPA receptors exist (Supplementary Fig. 13b).

We further examined whether upregulation of striatal FSI activity suppressed SWDs in *Stxbp1*+/− mice using the designer

receptors exclusively activated by designer drugs (DREADD) system. Because FSIs are parvalbumin (PV)-positive, we injected AAV-DIO-hM3D (Gq)-mCherry (Supplementary Fig. 14a) into the CPu of *Stxbp1*+/− mice crossed with PV-Cre driver mice, which led to the expression of DREADD (Gq) receptors in striatal FSIs in mice (Supplementary Fig. 14b). As expected, activation of FSIs by viral injection into the CPu (Fig. 6e, f), or intraperitoneal injection (Supplementary Fig. 14c) of the receptor agonist clozapine-N-oxide (CNO), both effectively suppressed the emergence of SWDs in *Stxbp1*+/− mice.

Finally, we recorded neuronal activity in the CPu of behaving *Stxbp1*+/− mice using extracellular electrodes. The recorded cells were classified into two neuron types, putative FSIs and MSNs (pFSIs and pMSNs, respectively), based on waveform

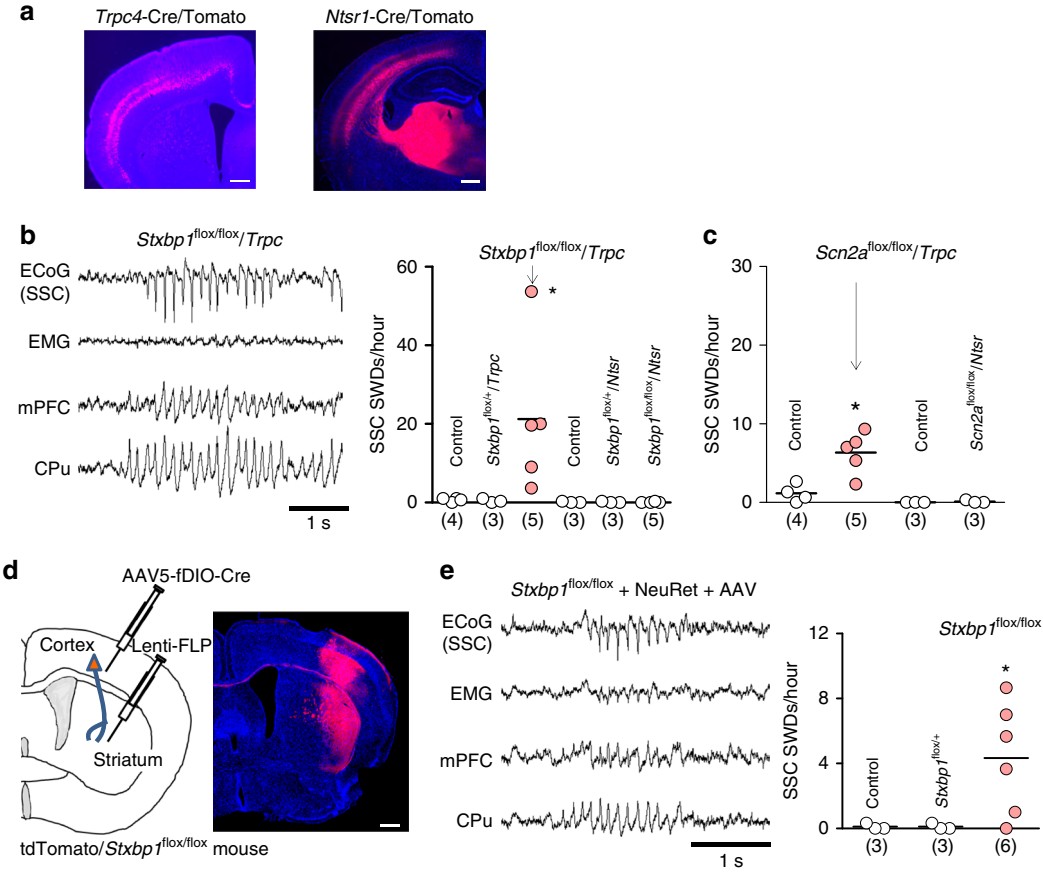

**Fig. 4** *Stxbp1* deletions in cortico-striatal but not cortico-thalamic projection neurons causes SWDs. **a** (Left) *Trpc4*-Cre dependent tdTomato (red) expression in neocortical layer 5 and the striatum (coronal section). (Right) *Ntsr1*-Cre dependent tdTomato expression (red) in neocortical layer 6 and the thalamus (coronal section). Scale bars: 500 μm. **b** *Stxbp1*$^{fl/fl}$/*Trpc*, but not *Stxbp1*$^{fl/fl}$/*Ntsr*, mice showed SWDs. (Left) Representative recordings. (Right) SSC SWDs numbers (3 h recording). Mann–Whitney $U$ test, *Stxbp1*$^{fl/fl}$/*Trpc* ($N = 5$) vs. control mice (3 *Stxbp1*$^{+/+}$/*Trpc*, 1 *Stxbp1*$^{fl/fl}$, $N = 4$), *$P = 0.0195$; *Stxbp1*$^{fl/fl}$/*Trpc* ($N = 5$) vs. *Stxbp1*$^{fl/+}$/*Trpc* ($N = 3$), *$P = 0.0357$; Kruskal–Wallis test, *$P = 0.0156$, Dunn's multiple comparison test, *Stxbp1*$^{fl/fl}$/*Trpc* ($N = 5$) vs. *Stxbp1*$^{fl/+}$/*Trpc* ($N = 3$), *$P < 0.05$. However, neither *Stxbp1*$^{fl/fl}$/*Ntsr* ($N = 5$) nor *Stxbp1*$^{fl/+}$/*Ntsr* ($N = 3$) mice increased SWDs, comparable to control mice (1 *Stxbp1*$^{+/+}$, 1 *Stxbp1*$^{+/+}$/*Ntsr*, 1 *Stxbp1*$^{fl/+}$, $N = 3$). Kruskal–Wallis test, $P = 0.3962$. **c** *Scn2a*$^{fl/fl}$/*Trpc*, but not *Scn2a*$^{fl/fl}$/*Ntsr*, mice showed SWDs (3 h recording). *Scn2a*$^{fl/fl}$/*Trpc* ($N = 5$) vs. *Scn2a*$^{+/+}$/*Trpc* ($N = 4$), Mann–Whitney test, *$P = 0.0317$. **d** Injected NeuRet vector (Lenti-FLP) in the striatum is retrogradely transported to the SSC, where FLP activates the Cre recombinase of the injected AAV5-EF1a-fDIO-Cre in the SSC. The Cre recombinase deletes floxed-*Stxbp1* and activates tdTomato genes (red). Scale bar: 500 μm. **e** SWDs appeared in mice with NeuRet-dependent *Stxbp1* deletion in cortico-striatal projection neurons (3 h recording). Unpaired $t$ test with Welch's correction, *Stxbp1*$^{fl/fl}$ ($N = 6$) vs. *Stxbp1*$^{fl/+}$ ($N = 3$), $t_5 = 3.027$, *$P = 0.0292$; *Stxbp1*$^{fl/fl}$ ($N = 6$) vs. *Stxbp1*$^{+/+}$ ($N = 3$), $t_5 = 3.027$, *$P = 0.0292$. Numbers of mice are shown in parentheses. *$P < 0.05$

characteristics (Fig. 7a, Supplementary Fig. 15a–d) and firing rates[44]. pFSIs displayed a narrower action potential spike width with a higher firing rate, whereas pMSNs displayed a broader spike width with a slow decay and lower firing rate[45]. Although FSIs supposedly constitute < 5% of total striatal neurons[38,42], we obtained ratios of pFSIs to the total number of the recorded neurons in *Stxbp1* mice of 15.9%, which was comparable to other studies (18.5%[45] or 10%[46]). This preferential detection of FSIs is presumably due to their characteristic short spike width and high firing rate. Notably, we frequently observed that pFSI activity dropped at the onset of cortical SWDs ($t = 0$) in behaving *Stxbp1*$^{+/-}$ mice (Fig. 7b, top row). Group data confirmed that pFSIs significantly decreased their neuronal activity at the onset of SWDs as compared with that observed at baseline, whereas pMSNs did not (Fig. 7c, d, Supplementary Fig. 15e–g). A significant increase in pMSN activity was not detected, possibly because only a subpopulation of MSNs might contribute to SWD generation. Furthermore, pFSIs occasionally displayed oscillatory spiking activity in phase with the oscillation observed in ECoG recordings during SWDs (Supplementary Fig. 15h).

To investigate whether impaired cortico-striatal excitatory transmission is also observed in animal models of typical absence epilepsy, we tested Genetic Absence Epilepsy Rats from Strasbourg (GAERS) rats, a well-established rat strain showing robust and spontaneous SWDs[47]. Occurrence frequency and duration of SWDs in GAERS rats (Fig. 8a) were larger than those in *Stxbp1*$^{+/-}$ mice (Fig. 1a, Supplementary Fig. 1d). Notably, microinjection of CX516 into the CPu of GAERS rats significantly reduced the number of SWDs (Fig. 8b), whereas NASPM microinjections increased the number of SWDs (Fig. 8c). These data might suggest the generality of our hypothesis that impaired excitatory inputs onto striatal FSIs leads to epilepsy (Fig. 9; see Discussion).

## Discussion

Thalamocortical circuits are widely recognized as the main generators of SWDs[19,20], whereas the basal ganglia (e.g., the striatum) have been proposed as merely modifying or suppressing SWDs[26–28,48–50]; however, a causal relationship between the basal ganglia and SWDs remains obscure. Studies that succeeded in

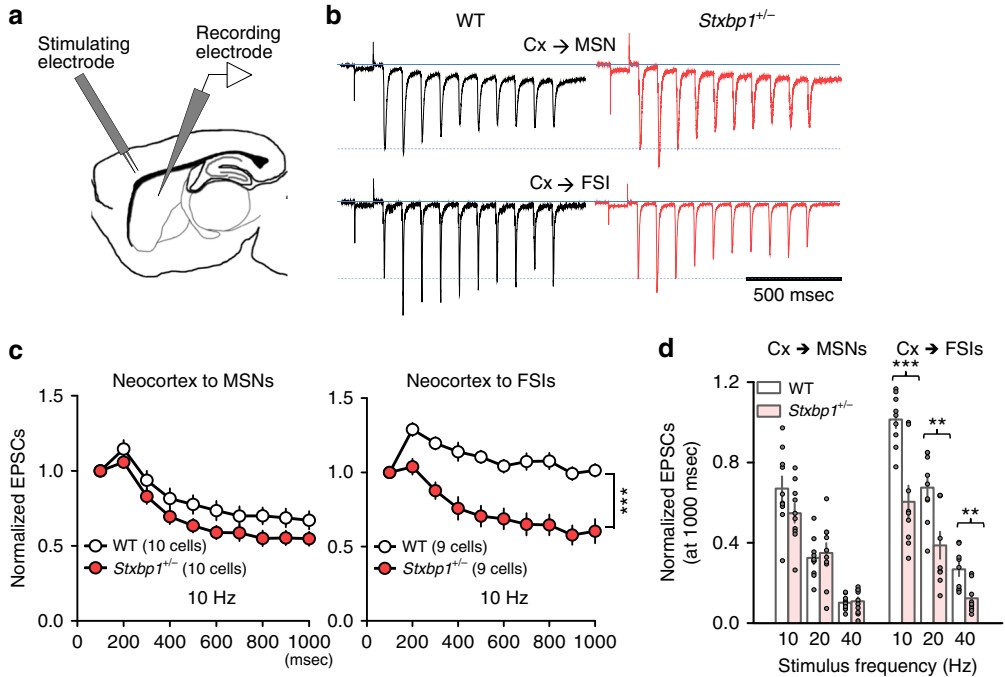

**Fig. 5** Impaired cortico-striatal FSI excitatory transmission in *Stxbp1*$^{+/−}$ mice. **a** Whole-cell recording configuration. **b** Representative traces of EPSCs in striatal MSNs and FSIs evoked by SSC electrical stimulations at 10 Hz. The initial EPSC responses are scaled to the dotted lines. **c** EPSCs showed faster rundown in FSIs (right), but not in MSNs (left), of *Stxbp1*$^{+/−}$ mice, compared to WT mice. 10 Hz stimulation [after 1000 ms stimulation, unpaired *t* test, WT (9 cells) vs. *Stxbp1*$^{+/−}$ (9 cells), FSI, $t_{16} = 4.46$, ***$P = 0.0004$]. **d** Steady-state levels of evoked EPSCs 1,000 ms after repetitive cortical stimulation ranging from 10 to 40 Hz. 10 MSNs and 9 FSIs, 19 slices from 9 WT mice; 10 MSNs and 9 FSIs, 19 slices from 9 *Stxbp1*$^{+/−}$ mice. Unpaired *t* test, WT vs. *Stxbp1*$^{+/−}$, MSN, 10 Hz: $t_{18} = 1.61$, $P = 0.128$; 20 Hz: $t_{18} = 0.4163$, $P = 0.6821$; 40 Hz: $t_{18} = 0.3123$, $P = 0.7584$. FSI, 10 Hz: $t_{16} = 4.46$, ***$P = 0.0004$; 20 Hz: $t_{16} = 3.277$, **$P = 0.0047$; 40 Hz: $t_{16} = 3.451$, **$P = 0.0033$. Mean ± SEM. *$P < 0.05$, **$P < 0.01$, ***$P < 0.001$

triggering SWDs by selective manipulations in control (non-epileptic) animals[23,51,52] were performed in the thalamus; however, such tests were not reported in the basal ganglia. In this study, we discovered that impairments of the cortico-striatal pathway caused SWDs. The causal role of the striatum in epileptogenesis sharply contrasts with the traditional concepts of the basal ganglia as merely a modulator. Furthermore, we unraveled a critical role for striatal FSIs in SWD generation. Our results provide strong evidence that a pathologic decrease in cortico-striatal excitatory transmission onto FSIs due to genetic mutations represents a causal driver of SWDs linked to epileptogenic phenotypes.

Gittis et al.[42] reported that the application of a calcium-permeable AMPA receptors blocker to the striatum decreased excitatory postsynaptic currents in cholinergic neurons (~45%) and FSIs (~73%), although the firing rates of only FSIs, and not cholinergic neurons, were selectively decreased. However, we still cannot exclude the possibility that epileptic activity caused by NASPM was partially mediated by cholinergic interneurons.

Pharmacological suppression[42], cell ablation[53], and optogenetic suppression of striatal FSIs[46] have been performed in mice. Neither SWDs nor epileptic seizures were described in these reports; however, Klaus and Plenz[43] found SWD-like cortical LFP changes followed by pharmacological FSIs inhibition, which is similar to our observation. In particular, non-convulsive seizures such as absence seizure are very difficult to detect without electroencephalographic recordings. Even *Stxbp1*$^{+/−}$ mice, *Scn2a*$^{+/−}$ mice, and GAERS rats showing frequent SWDs appear outwardly normal. It is likely that in previous studies not designed to detect epilepsies, SWDs, or other epileptic brain activity, these may have been overlooked.

In accordance with this finding, we propose a novel cortico-striato-thalamic neural circuit for epilepsy that traverses the basal ganglia via an indirect pathway (Fig. 9a, b). In this model, impaired cortico-striatal excitatory neurotransmission diminishes FSI activity, which disinhibits MSNs in the CPu. Activated MSNs then sequentially over-suppress the globus pallidus externus (GPe), resulting in disinhibition of the subthalamic nucleus (STN), activation of the globus pallidus internus/substantia nigra pars reticulata (GPi/SNr), and over-suppression of the thalamus. Consequently, the conventional model of hyperpolarized thalamic relay neurons produces rebound firing by de-inactivating Cav3.1 T-type $Ca^{2+}$ channels[29] to further generates SWD and seizures.

Cortico-striatal inputs predominantly activate enkephalin-positive MSNs in the indirect pathway[54], suggesting MSN-GPe as a major route for the seizures. Previous results obtained using the GAERS absence epilepsy rat model are consistent with our pro-epileptic cortico-striatal circuit. Specifically, blockade of GABA inhibition in the GPe[27] or glutamate receptors in the SNr[26,28], and muscimol inactivation of the STN[26] or SNr[26,55] suppressed SWDs, whereas blockade of GABA$_A$ receptors in the SNr aggravates SWDs[27]. Moreover, STN membrane excitability was enhanced in an absence epilepsy mouse model[50]. However, because there are reciprocal connections between direct and indirect pathway neurons, it also remains to be determined how the disinhibition of FSIs preferentially affects SWDs via the indirect pathway. Direct/indirect pathway-selective manipulation and a more detailed analysis of neuronal interactions among the cortex, basal ganglia, and thalamus are required in future studies to confirm our model.

In *Stxbp1*$^{+/−}$ mice, in vivo FSI activity decreases before or at the occurrence of SWDs, leading to disinhibition of MSNs. Potential mechanisms might include a faster rundown of excitatory transmission onto FSIs but not MSNs in *Stxbp1*$^{+/−}$ mice or a higher responsiveness of FSIs to cortical inputs relative

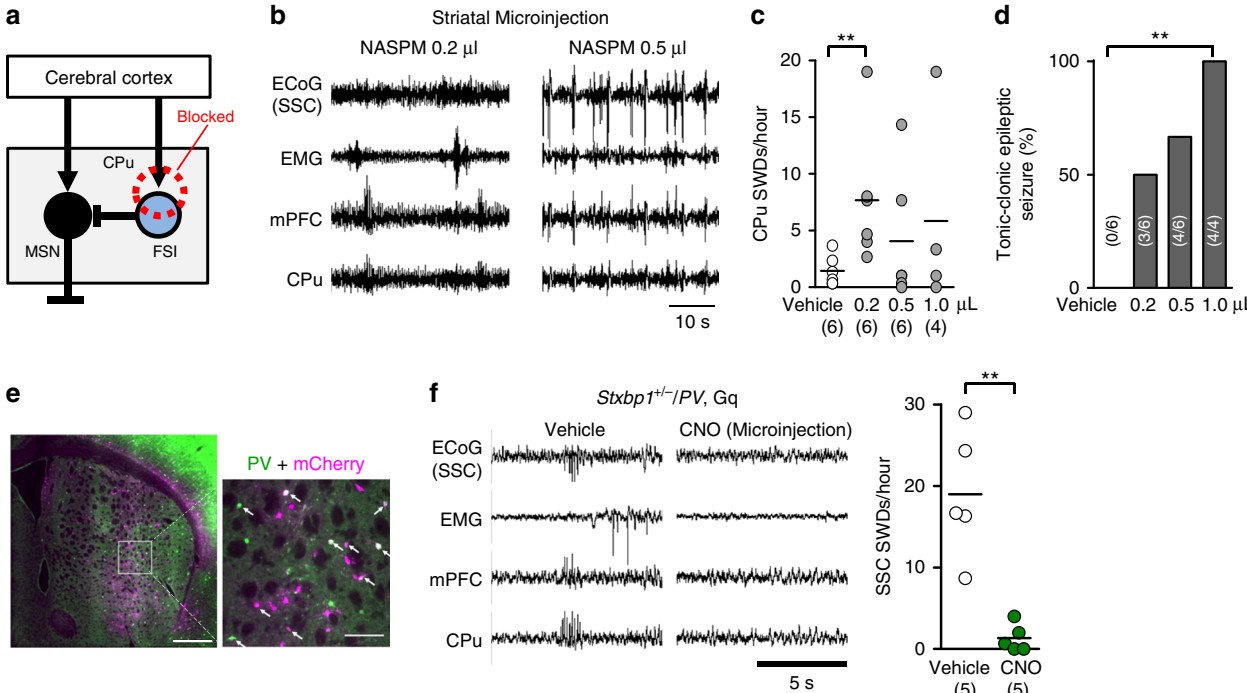

**Fig. 6** Blockade of cortico-striatal FSI excitatory transmission causes SWDs and activation of FSI suppresses SWDs. **a** Local blockade of excitatory inputs to striatal FSIs by NASPM. **b** SSC-ECoG and LFP recordings at low (0.2 μl) and high (0.5 μl) doses of NASPM microinjection into the CPu of a WT mouse. SWDs in the mPFC and CPu (0.2 μl) (left) and convulsive epileptic discharges in the SSC, mPFC, and CPu (0.5 μl) (right). **c** Dose-dependent striatal SWDs in WT mice, caused by NASPM injections (3 h recording after injection). Vehicle ($N = 6$) vs. low NASPM ($N = 6$), Mann–Whitney $U$ test, **$P = 0.0081$. **d** Dose-dependent convulsive seizures caused by NASPM (3 h recording after injection). Mouse numbers are shown in parenthesis. Vehicle ($N = 6$) vs. NASPM (1. 0 μl, $N = 4$), Fisher's exact test, **$P = 0.0048$. **e** The majority of PV (green)-positive FSIs in the CPu expressed mCherry (magenta) after AAV injections, indicating expression of the DREADD receptors (arrows: double positive cells, including those with less dense PV signals). Coronal section, scale bars 500 μm (left), 100 μm (right). **f** SWDs were suppressed by CNO microinjection into the CPu of $Stxbp1^{+/-}/PV$ Gq mice. Number of SWDs (right, 3 h recording). $Stxbp1^{+/-}/PV$ Gq, vehicle vs. CNO, Mann–Whitney $U$ test, **$P = 0.0079$. Mean or Mean ± SEM. **$P < 0.01$

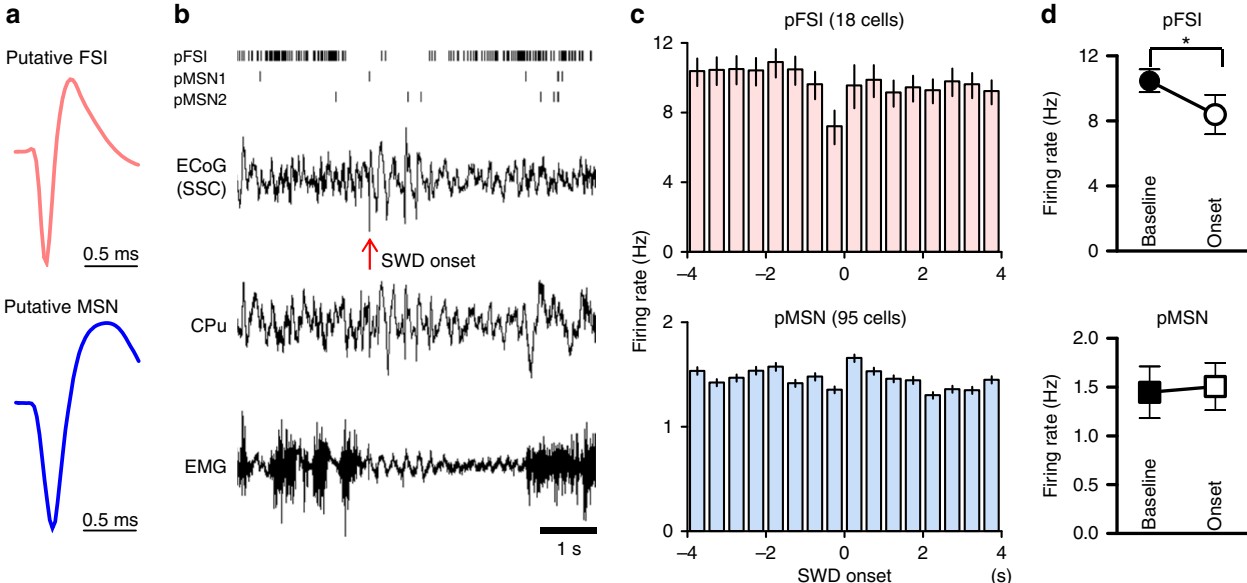

**Fig. 7** Temporal down-regulation of striatal FSIs at SWD onset in $Stxbp1^{+/-}$ mice. **a** Averaged spike waveforms of single units from a pFSI and a pMSN (FSI, 52,478 spikes; MSN 1,369 spikes). **b** Representative single unit activities of one pFSI and two pMSNs (raster plot, top), simultaneously recorded SSC-ECoG, EMG, and CPu LFP around the onset of cortical SWD (arrow). **c** Averaged peri-event time histograms (500 ms bins) of pFSI (18 cells) and pMSNs (95 cells) from 10 $Stxbp1^{+/-}$ mice. The timings of the SWD onset were aligned at 0 s. **d** pFSIs activity at the onset of SWDs ($-0.5$ to $+0.5$ s) significantly decreased compared to baseline ($-5$ to $-4$ s), while pMSNs did not. pFSIs: baseline vs. onset, Mann–Whitney $U$ test, *$P = 0.0279$, pMSNs: $P = 0.5831$. Mean ± SEM. *$P < 0.05$

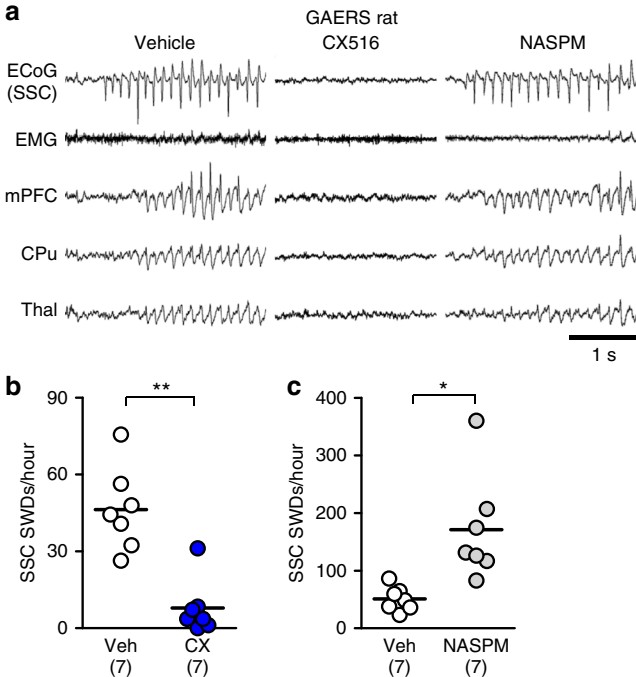

**Fig. 8** Facilitation and blockade of cortico-striatal excitatory transmission suppresses and aggravates SWDs in GAERS rats. **a** Representative ECoG or LFP recordings in GAERS rat with vehicle, CX516 and NASPM injections. **b, c** Microinjection of CX516 into the striatum (1.5 µl, bilateral) suppressed SWDs (**b**), while NASPM injections into the striatum (1.5 µl, bilateral) aggravates SWDs in GAERS rats (**c**). Vehicle ($N = 7$) vs. CX516 ($N = 7$), unpaired $t$ test with Welch's correction, number of SWDs for 3000 sec after injection: $t_{10} = 5.218$, **$P = 0.0004$; total duration of SWDs: 401.5 ± 67.0 sec (vehicle), 24.6 ± 17.6 sec (CX516), $t_6 = 5.439$, **$P = 0.0016$. **b**, Vehicle ($N = 7$) vs. NASPM ($N = 7$), unpaired $t$ test with Welch's correction, number of SWDs for 2000 sec after injection: $t_6 = 3.356$, *$P = 0.0153$; total duration of SWDs: 306.2 ± 41.2 sec (vehicle), 488.8 ± 68.8 sec (NASPM), $t_9 = 2.277$, *$P = 0.0488$. *$P < 0.05$, ***$P < 0.005$

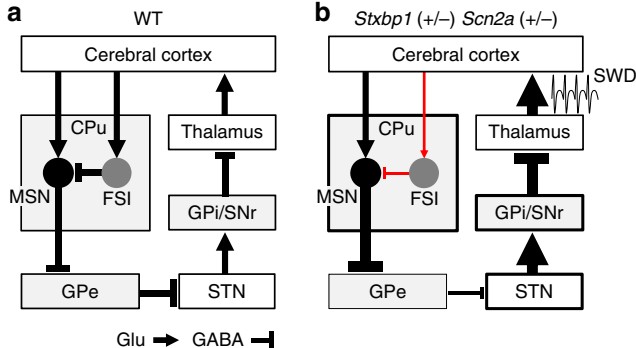

**Fig. 9** A striatal circuit model for epilepsy in $Stxbp1^{+/-}$ and $Scn2a^{+/-}$ mice. **a** Canonical neural circuit of the indirect pathway in WT mice. **b** Hypothetical circuit for absence seizures in $Stxbp1^{+/-}$ and $Scn2a^{+/-}$ mice. Thicker and thinner lines indicate increased and decreased neural transmissions, respectively

to that of MSNs[37,54] (Fig. 9b). Additionally, we observed that the effects of $Scn2a$ haploinsufficiency on cortico-striatal transmission were minor (Supplementary Fig. 11d, e). Because $Scn2a$ haploinsufficiency results in the broadening of action potentials[15], this could lead to excessive glutamate release followed by a depletion upon repetitive activity. Reduced glutamate

transmission in the cortico-striatal pathway is also a possible mechanism in $Scn2a^{+/-}$ mice.

Our results including the reproductions of SWDs by $Trpc4$-Cre- (cortico-striatal neuron-specific) but not $Ntsr1$-Cre- (thalamocortical neuron-specific) dependent $Stxbp1$ or $Scn2a$ deletions or by cortico-striatal projection neurons-specific deletion of $Stxbp1$ (NeuRet) and the induction of SWDs by brief electrical stimulation of the CPu in $Stxbp1$ mice, support the cortico-striatal pathway as the initial site causally responsible for the seizures and subsequent activation of thalamo-cortical circuits at least in $Stxbp1^{+/-}$ and $Scn2a^{+/-}$ mice, although the cortico-striatal-thalamic loop and cortico-thalamic loop are not mutually exclusive and they might influence SWDs cooperatively.

$STXBP1$ and $SCN2A$ mutations have been described in patients with epileptic encephalopathies who show myoclonic, atonic, and atypical absence seizures[1–5,7,8], although our model (Fig. 9b) might also cover typical absence epilepsy, as described. Behavioral quiescence, myoclonia, dyskinesia-like tonic or clonic seizures, and tonic-clonic seizures appeared sequentially in a NASPM dose-dependent manner, which was consistent with the presence of dyskinesia in mice following pharmacological suppression of striatal FSIs[42]. Choreoathetoid movements have also been described in some patients with $STXBP1$[6] and $SCN2A$[56] mutations. These data suggest that these features are derived from overlapping pathological circuits, and that impaired cortico-striatal excitatory transmission contributes not only to absence (i.e., non-convulsive) seizures, but also to convulsive seizures.

In summary, this study fills an important gap in the seizure literature by addressing how impaired excitatory transmission is frequently observed in models of absence epilepsy[19,20,57]. Future studies will address this pro-epileptogenic striatal neural circuit in the context of excitatory balance in neurodevelopmental disorders[58,59] and why autism and epilepsy have a high rate of association[60].

## Methods

**Animals**. All animal experimental protocols were approved by the Animal Experiment Committee of the RIKEN Center for Brain Science. Mice and rats were handled in accordance with the guidelines of the RIKEN Center for Brain Science Animal Experiment Committee. Food and water were available ad libitum, and cages (of less than 5 animals) were kept at 23 °C and 55% humidity on a 12-h/12-h light/dark cycle, with the lights off at 20:00. Genetic absence epilepsy rat from Strasbourg (GAERS/Mave, NRPB Rat No. 0285) were obtained from the National BioResource Project - Rat, Kyoto University (Kyoto, Japan). Both sexes, over 16 weeks of age.

**$Stxbp1$ and $Scn2a$ conditional knockout mice**. $Stxbp1$ floxed mice, which have exon 3 flanked by loxP sites, were maintained on a C57BL/6 N background[11]. To generate $Stxbp1$ conditional knockout mice, heterozygous $Stxbp1$ floxed ($Stxbp1^{fl/+}$) or homozygous $Stxbp1$ floxed ($Stxbp1^{fl/fl}$) mice were mated with Cre transgenic mice. The Empty spiracles homolog 1 ($Emx1$)-Cre knock-in mice[11,31,32], vesicular GABA transporter ($Vgat$)-Cre BAC transgenic mice[11,32], short transient receptor potential channel 4 ($Trpc4$)-Cre transgenic mice[34], neurotensin receptor 1 ($Ntsr1$)-Cre transgenic mice (Tg($Ntsr1$-cre)GN220Gsat/Mmcd, GENSAT, MMRRC), parvalbumin (PV)-Cre transgenic mouse[61], and loxP flanked transcription terminator cassette CAG promotor driven tdTomato transgenic (B6.Cg-$Gt(ROSA)26Sor^{tm14(CAG-tdTomato)Hze}$/J, Stock No: 007914, The Jackson Laboratory, USA) mice were all maintained on a C57BL/6 J background. $Scn2a$ floxed mice, which have exon 2 flanked by loxP sites, were maintained on a C57BL/6 J background[15]. To generate $Scn2a$ conditional knockout mice, Heterozygous $Scn2a$ floxed ($Scn2a^{fl/+}$) or homozygous $Scn2a$ floxed ($Scn2a^{fl/fl}$) mice were mated with Cre transgenic mice.

**Somatosensory cortex (SSC) electrocorticogram (ECoG) and electromyogram (EMG) recordings**. Adult mice (>8 weeks, both sexes) were used in this study. ECoG was used to directly monitor cortical activity using stainless steel screw electrodes embedded in the skull. Stainless steel screws (1.1 mm diameter) serving as ECoG electrodes were placed over the bilateral somatosensory cortex (±1.5 mm lateral to midline, 1.0 mm posterior to bregma) under 1–1.5% isoflurane anesthesia or 1.5% halothane anesthesia with $N_2O{:}O_2$ (3:2) ventilation. A reference screw

electrode was implanted on the cerebellum (at midline, 1.5 mm posterior to lambda). A stainless steel wire (100 μm) bipolar electrode was inserted in the cervical region of the trapezius muscle for EMG. Beginning at least one week after surgery, recordings were performed for a week, sampled at 256 Hz and then analyzed off-line (SleepSign, Kissay, Japan). Each animal's behavior was continuously monitored using an infrared camera.

**Simultaneous ECoG and Local field potential (LFP) recordings from behaving mice.** Adult mice (>8 weeks, both sexes) were used. Stainless steel screws (1.1 mm diameter) serving as ECoG electrodes were placed over the right somatosensory cortex (1.0 mm posterior to bregma, 1.5 mm to the midline) under 1–1.5 % iso-flurane anesthesia, or Nembutal (50 mg/kg of body weight) and 1.5% halothane anesthesia with $N_2O:O_2$ (3:2) ventilation[11]. A stainless-steel screw, serving as both a reference and ground electrode, was placed on the cerebellum. A stainless-steel wire monopolar electrode was inserted in the cervical region of the trapezius muscle for EMG. To record monopolar LFP recordings of brain regions, insulated stainless steel wires (200-μm diameter) with beveled tips were stereotaxically implanted contralateral to the ECoG electrode, according to the following coordinates (anterior-posterior, medial-lateral, and depth from the cortical surface, mm): medial prefrontal cortex (1.9, 0.3, 1.4), caudate-putamen (0.0, 2.4, 2.5), basolateral amygdala (−1.4, 2.9, 3.7), ventroposterior thalamus (−1.8, 1.5, 3.2), hippocampus CA1 region (−2.5, 2.2, 1.1), and visual cortex binocular zone (−3.4, 3.0, 0.4). Contacts between the electrode and brain surface were covered with a small amount of petroleum jelly and secured with dental acrylic. An antibiotic (ampicillin) was used during surgery.

Following at least one week of recovery from the implant surgery, the animals were tethered to a 16-channel commutator (Plexon, Dallas, TX) and allowed to move freely during recording in an electrically shielded cage with food and water available ad libitum. A light/dark (12:12) cycle was kept in the recording chamber using a dim light. LFPs (filtered 0.7–170 Hz, 1 kHz sampling) were recorded using the MAP data acquisition system (Plexon, Dallas, TX, USA) and analyzed off-line using a software program (NeuroExplorer, Nex Technology, Madison, AL, USA). Power spectrum analysis of ECoG was also determined (0–20 Hz, 0.078125 Hz bin) by NeuroExplorer.

**Discrimination of behavioral states.** Brain states were scored manually for each 4 s epoch based on SSC ECoG and EMG using SleepSign software (Kissei, Japan). The wakeful state is characterized by low-amplitude desynchronized activity with high EMG activity; slow-wave sleep (Non-REM: NREM) sleep is characterized by high-amplitude, slow wave activity (1–4 Hz dominant) ECoG with low EMG activity; and REM sleep is associated with theta (around 8 Hz dominant) ECoG with low EMG activity[62,63].

**Definition of Spike-and-Wave discharges (SWDs).** An SWD was defined as at least three spike-and-waves consisting of negative (for SSC-ECoG, LFP) or positive (for LFPs in the mPFC, CPu, Amg, Thal, CA1, or VC) spikes, 6–10 Hz (SSC) or 3–10 Hz (mPFC, CPu, Amg, Thal, CA1, and VC) in frequency, and with spike amplitude over 200% of baseline activity at wakefulness. Isolated SWDs with lower frequency occasionally occur in the mPFC, CPu, Amg, Thal, CA1, or VC. Manual scoring was performed by scientists blinded to the genotypes and experimental treatments. We observed positive polyspikes in the ECoG, which was also quantified (Supplementary Fig. 1c).

**Quantification of behavioral quiescence and twitches/jumps.** Quantification of behavioral quiescence during SWDs was based on EMG activity during the active waking period. When EMG activity (amplitude/intensity) stopped, dropped, or decreased during SWD compared to EMG activity (4 s) just before the SWD, by visual inspection, it was counted as behavioral quiescence. Twenty SWDs were randomly sampled from each mouse during waking with active EMG. Using video monitoring, twitches or jumps[14] from the sleep state were identified (6 h during light phase) in mice with or without ECoG recording.

**Microinjection and systemic administration of drugs.** A guide cannula (C313G, 7 mm in length) and internal dummy cannula (C313DC, 8 mm in length, Plastics One, VA, USA) were implanted into mice (>8 weeks), in combination with the ECoG/EMG and LFP recordings (mPFC and CPu). The tip of the injection cannula (C313I: 360-μm diameter, 8-mm length) extrudes 1 mm from the end of the guide cannula. The tip of injection cannula was targeted to either the SSC (−1.0, 1.5, 0.4), CPu (0.0, 2.4, 2.5), mPFC (1.9, 0.3, 1.4), mediodorsal thalamus (−1.5, 0.4, 3.0), ventroposterior thalamus (−1.8, 1.5, 3.0), or CA1 (−2.5, 2.2, 1.1), unilaterally, contralateral to the ECoG recording site.

Muscimol (1 mM, Wako), bicuculline methiodide (10 mM, Tocris) or CX516 (100 mg/ml, Sigma) was dissolved in 0.2 μl of vehicle solution (saline) and slowly (>2 min) injected unilaterally with a micro-syringe via a polyethylene tube connected to the injection cannula. The injection cannula was withdrawn 30 s after injection. During the injection, the mouse was lightly anesthetized with isoflurane, or halothane and $N_2O:O_2$. In the muscimol experiments, SWDs were quantified for 3 h after microinjections. In the CX516 experiments, SWDs were counted in the first 50 min after injections, multiplied by three and then divided (normalized) by

the total number of SWDs counted for 150 min following vehicle injections, due to inter-individual variability of SWDs. 1-Naphthyl acetyl spermine (NASPM) (5 mM in saline, Tocris) was microinjected bilaterally, at doses of 0.2, 0.5, or 1.0 μl. SWDs were quantified for 3 h after NASPM microinjections. Vehicle or drug was given to the same mice, with at least a 3-day interval between injections.

For systemic administration, ethosuximide (33.3 mg/ml in saline, 200 mg/kg body weight, Toronto Research Chemicals) or vehicle (13.3 μl/g body weight of saline, Otsuka) was intraperitoneally injected. SWDs were quantified for 2.5 h after injections. In CX516 experiments, CX516 (10 mg/ml in saline, 130 mg/kg body weight, Sigma) or vehicle (13.3 μl/g body weight of saline, Otsuka) was intraperitoneally injected. SWDs were quantified during the first 60 min (0–60 min) and the second 60 min (60–120 min) after the injections. Vehicle or drug was given to the same mice, with at least a 3-day interval between injections.

**In vivo electrical stimulation.** Brief monophasic electrical stimulation (duration 100–200 μs, intensity 100–200 μA, 0.25 Hz) was applied to the CPu of adult $Stxbp1^{+/−}$ mice through twisted stainless-steel wire (100 μm), bipolar electrodes. Stimulating electrodes and a CPu LFP recording electrode were bundled, with the tips of the stimulating electrodes separated by about 300 μm. Stimulus trains and constant currents were generated by a stimulator (SEN-8203, Nihon Kohden, Tokyo, Japan) and by an isolator (SS-202J, Nihon Kohden), respectively. ECoG/EMG and LFPs (of the mPFC, CPu, and ventroposterior thalamus) were monitored simultaneously.

**In vivo microdialysis.** To measure extracellular glutamate and GABA in vivo, a linear microdialysis probe (membrane length 2 mm, cut off 50,000 MW, Eicom, Kyoto, Japan) was implanted into the CPu. One week after surgery, animals were tethered to the microdialysis system via a liquid swivel and allowed to move freely in the daytime. Ringer's solution (147 mM NaCl, 4.0 mM KCl, 4.5 mM $CaCl_2$) was perfused (2.0 μl/min) and samples were collected every 20 min using a fraction collector (CMA 140). After an equilibration phase (120 min), six fractions were sampled as baseline, then the Ringer's solution was changed to a high $K^+$ solution (100 mM KCl, 51 mM NaCl, 2.3 mM $CaCl_2$), using a liquid switch, for 60 min, and then returned to Ringer's solution for 60 min. Dialysates in aliquots were immediately stored at −80 °C until measurement. Glutamate and GABA in dialysates were quantified using high-performance liquid chromatography (Eicom) with fluorescence detection (excitation, 340 nm; emission, 440 nm). Amino acids, after o-phthalaldehyde derivatization, were separated on a Cosmosil 5C18 reversed-phase column (at 36 °C, Nacalai Tesque, Japan) with the column-switching technique (EAS-20s, Eicom). The components of the mobile phases were as follows: 88% 0.1 M citrate buffer/0.2 M phosphate buffer (pH 6.2) and 12% acetonitrile, for glutamate; 75% 0.1 M citrate buffer/0.2 M phosphate buffer (pH 5.4) and 25% acetonitrile, for GABA[64].

**Neuron-specific retrograde transport vector (NeuRet) system.** For construction of flippase (FLP) expression lentiviral vectors (Supplementary Fig. 9a, b), an FLP recombinase gene fragment and the pCL20c Mp + vector were digested with BsrGI and NotI and then ligated. The FLPe gene was amplified using the following primers: 5′-TAGGATCCTGTACAGCCACCATGGCTCCCAAGAAGAAGAG-3′ and 5′-GATCGCGGCCGCTTATATGCGTCTATTTATGTAGGATG-3′.

Subsequently, to obtain a construct with a fluorescence expression marker, an IRES-EGFP fragment was inserted into the NotI site, 3′ of the FLPe gene, that was filled by the PrimeSTAR® HS DNA polymerase (TAKARA). The IRES-EGFP fragment was amplified from the pIRES2-EGFP vector (Clontech) by PCR using the following primers:
IRES2_5′-end: 5′-GGATCCGCCCCTCTCCCT-3′,
EGFP_3′-end: 5′-GCTTTACTTGTACAGCTCGTCC-3′.
For construction of the FLP-dependent Cre expression AAV vectors, we designed an inverted Cre gene sequence flanked by Frt and F5 sites, based on the sequence data of the Addgene plasmids #55641 (fDIO) and #24593 (Cre): 5′-Frt-F5-(3′-Cre recombinase-5′)-Ftr-F5-3′. The fDIO-Cre fragment was chemically synthesized by GeneArt® (Thermo Fisher Scientific). The GeneArt® product and pAAV-CAGGS-MCS-WPRE were digested by BamHI and EcoRV and then the fDIO-Cre fragment and the pAAV vector were ligated. Packaging of the plasmid vectors into lentivirus or AAV, and purification and quantification of the viruses were performed[36] at the section for viral vector development of the National Institute of Physiological Sciences.

For the cell culture assay, 3 μg of plasmid DNA was transfected into HEK293FT cells (Thermo Fisher Scientific, R70007) by lipofection, using Lipofectamine LTX with Plus reagent (Thermo Fisher Scientific #15338100). HEK293FT cells were cultured in Poly-L-lysine coated 6-well culture plate, with DMEM + 10% FBS.

$Stxbp1^{fl/fl}$ mice were crossed with tdTomato transgenic mice to obtain $Stxbp1^{fl/fl}$/tdTomato mice which expresses a red fluorescent protein (tdTomato) by Cre-mediated recombination. Retrograde lentiviral (NeuRet) vectors (titer: 2.1–6.1 × $10^{11}$ copies/ml) were injected into the CPu of adult $Stxbp1^{fl/fl}$, $Stxbp1^{fl/+}$ or $Stxbp1^{+/+}$/tdTomato mice (>2 months of age) with the following stereotaxic coordinates: (1.2, ± 2.0, 2.0 mm, virus 1.0 μl), (0.5, ± 2.5, 2.5 mm, 1.4 μl), and (−0.25, ± 2.7, 2.7 mm, 1.4 μl), using a glass pipette (diameter < 30 μm) with a

microinjector (IMS-20, Narishige, Japan) (0.1 μl/min). One week after NeuRet vector injections, the mice were injected with the AAV5 vector (CAG promotor fDIO-Cre, titer $4.5 \times 10^9$ viral genomes/μl) into the cerebral cortex, as follows: (1.0, ± 1.5, 0.8 mm, 0.3 μl), (1.0, ± 3.2, 1.0 mm, 0.3 μl), (−0.5, ± 1.5, 0.8 mm, 0.3 μl), (−0.5, ± 3.2, 1.0 mm, 0.3 μl), (−1.5, ± 1.5, 0.6 mm, 0.3 μl), and (−1.5, ± 3.2, 1.0 mm, 0.3 μl). One week after AAV injection, the electrodes were implanted. At least 4 weeks following the final virus injections, we began ECoG and LFPs recordings.

**DREADD system.** AAV vectors were produced[36] at the section of viral vector development of the National Institute of Physiological Sciences, packaging plasmids of pAAV-EF1a-DIO-hMD3(Gq)-mCherry (Addgen plasmid #50460, a gift from Bryan Roth) or pAAV-EF1a-DIO-mCherry (Addgene plasmid #47636, a gift from Brandon Harvey) into AAV5.

Parvalbumin-Cre transgenic (PV-Cre) mice express Cre recombinase in the majority of parvalbumin -positive cells[32,61]. $Stxbp1^{+/−}$ mice were crossed with PV-Cre mice to obtain $Stxbp1^{+/−}$/ PV-Cre mice. AAV5 vectors for DREADD were injected into the striatum of adult $Stxbp1^{+/−}$/ PV-Cre or $Stxbp1^{+/+}$/ PV-Cre mice (>2 months of age), with the following stereotaxic coordinates: (0.7, ± 2.0, 2.2 mm, virus 0.8 μl), (0.0, ± 2.0, 2.2 mm, 0.8 μl), using a glass pipette (diameter < 30 μm) with a microinjector (IMS-20, Narishige, Japan) (0.1 μl/min). One week after AAV vector injections, the mice were implanted with electrodes for ECoG, and the mPFC and CPu with bilateral cannulae aimed at the CPu for clozapine-N-oxide (CNO) microinjection. At least 4 weeks after the AAV injections, we commenced ECoG and LFPs recordings. The virus titers were as follows: AAV5-EF1a-DIO-hM3D (Gq)-mCherry: $8.8 \times 10^9$ viral genomes/μl, and AAV5-EF1a-DIO-mCherry: $9.5 \times 10^9$ viral genomes/μl. Clozapine-N-oxide (CNO) (2.9 mM in 0.5% DMSO saline, Tocris) was dissolved in 0.2 μl of vehicle solution (0.5% DMSO saline) and slowly (>2 min) injected bilaterally with a micro-syringe via a polyethylene tube connected to the injection cannula. The injection cannula was withdrawn 30 s after the injection.

**Immunohistochemistry and fluorescence histochemistry.** Mice were deeply anesthetized and perfused transcardially with 4% paraformaldehyde in phosphate-buffered saline (PBS)[32]. The brains were then removed, post-fixed in 4% paraformaldehyde in PBS, and cryoprotected. Frozen brain sections (30 μm) were incubated with rabbit anti-parvalbumin antibody (1:5000) (PC255L; Calbiochem, San Diego, CA, USA), and then in secondary antibody (donkey anti-rabbit IgG, Alexa Fluor 488-conjugated, 1:1000; Thermo Fisher Scientific). Sections were washed with PBS containing 4′6-diamidino-2-phenylindole (DAPI; Nacalai Tesque, Kyoto, Japan) and mounted with ProLongGold (Thermo Fisher Scientific). For fluorescence histochemistry of NeuRet brain samples, frozen brain sections (50 μm) were incubated with PBS containing DAPI and mounted with ProLongGold. Images were captured and processed using a fluorescence microscope (BZ-X710; Keyence, Osaka, Japan) and image-analyzing software (BZ-X analyzer; Keyence, Osaka, Japan).

Upon completion of the experiment using ECoG and LFP recordings, the positions of the electrodes and cannulas were verified histologically using hematoxylin and eosin staining (30-μm brain sections). Photomicrographs were acquired with light microscopes, BZ-8000 (Keyence, Osaka, Japan).

**In vitro slice electrophysiology.** Sagittal brain slices (350-μm thickness) collected from the mice at postnatal day 24 to 32 (P24–32) were cut on a vibratome (PRO 7, DOSAKA EM, Kyoto, Japan) under isoflurane anesthesia and incubated for at least 1 h in warmed (34 °C) and equilibrated (95% $O_2$, 5% $CO_2$) artificial cerebrospinal fluid (ACSF) containing (in mM) 125 NaCl, 2.5 KCl, 1 $MgSO_4$, 2 $CaCl_2$, 1.25 $NaH_2PO_4$, 26 $NaHCO_3$, 11 glucose, 3 Na-pyruvate, and 1 Na-L-ascorbic acid, before recording. Whole cell patch-clamp recordings were made from MSNs and FSIs in the striatum.

Patch pipettes (4–6 MΩ) were pulled from borosilicate glass on a puller (DMZ-UNIVERSAL PULLER, Zeitz Instruments, Germany) and filled with an internal solution containing (in mM) 128 K-gluconate, 8 KCl, 2 NaCl, 0.2 EGTA, 20 HEPES, 4 MgATP, 0.3 $Na_3GTP$, 14 phosphocreatine, and 0.1 Alexa Fluor 594 hydrazide. Since neurotransmitter release critically depends on temperature, we kept brain slices at 30–31 °C to obtain stable recordings. Data were obtained with a MultiClamp 700B amplifier, filtered (1 kHz), digitized (10 kHz), stored, and analyzed using pCLAMP 10.3 software (Molecular Devices, Foster city, CA). Series resistance throughout the recordings under current-clamp mode was monitored by applying a hyperpolarizing square pulse (100 pA for 100 ms) and compensated with a bridge balance circuit in the amplifier. Under voltage-clamp mode, series resistance was monitored with a 2-mV hyperpolarizing square pulse but not compensated. Data were discarded when series resistance exceeded > 20 MΩ or varied by more than ± 10%.

To analyze the properties of presynaptic glutamate release onto cortico-striatal synapses, a bipolar electrode was placed in layer 5/6 adjacent to the striatum, and 10, 20, and 40 Hz stimulations were applied for 1 s, four times, at 30-s intervals. Asynchronous mini-EPSCs were analyzed during an 800-ms period after the end of the paired synchronous response under extracellular $Ca^{2+}$ (2 mM) and $Sr^{2+}$ (4 mM) conditions.

Following in vitro slice recording, recorded cells were filled with biocytin (diluted 1% in internal solution for patch-clamp recording; Sigma B1758). Sections were then post-fixed in 4% PFA for 48 h. The tissue was then permeabilized in 0.3% Triton X-100 in PBS, blocked in BlockAce (4% in PBS- 1% Tween20 for 1 h; DS Pharma Biomedical UK-B80) and incubated overnight at 4 °C with a rabbit anti-PV (1/5,000; calbiochem PC255L) in 4% BlockAce. The sections were then incubated overnight with a mix containing an Alexa 488 labelled anti-rabbit antibody (1/1,000; Invitrogen A21206) and Alexa 594 labelled streptavidin (1/1,000; Molecular Probes S11227). Samples were then counterstained in DAPI, transferred on glass slides and glass covers were mounted in ProLong Gold (Invitrogen P36934). Images were acquired using a BZ-X710 fluorescence microscope (Keyence).

**Microinjection of drugs into the striatum of GAERS rats.** Adult GAERS rats (>16 weeks, both sexes) were also subjected to CPu microinjection experiments. Similarly to mice, stainless steel screws (1.1 mm diameter) serving as ECoG electrodes were placed over the right somatosensory cortex (2.7 mm posterior to bregma, 4.0 mm to the midline) under 2% isoflurane anesthesia (initial 4%). A stainless-steel screw, serving as both a reference and ground electrode, was placed on the cerebellum. A stainless-steel wire monopolar electrode was inserted in the cervical region of the trapezius muscle for EMG. To record monopolar LFP recordings of brain regions, insulated stainless steel wires (200-μm diameter) were stereotaxically implanted contralateral to the ECoG electrode, according to the following coordinates (anterior-posterior, medial-lateral, and depth from the cortical surface, mm): medial prefrontal cortex (3.2, 0.5, 3.0), caudate-putamen (0.0, 3.5, 4.6), ventroposterior thalamus (−3.2, 2.8, 5.3). The tip of the injection cannula (C313I: 360-μm diameter, 8-mm length) extrudes 1 mm from the end of the guide cannula. The tip of injection cannula was targeted to the CPu (0.0, 3.5, 4.6) bilaterally.

Following at least one week of recovery from the implant surgery, microinjections of drugs in rats were performed. CX516 (100 mg/ml, Sigma) or NASPM (8 mM in saline, Tocris) was dissolved in 1.5 μl of vehicle solution (saline) and slowly (>2 min) injected bilaterally. During the injection, the rat was anesthetized with isoflurane. Vehicle, CX516 or NASPM was given to the same GAERS rats.

**Single units recording from striatum of behaving mice.** To acquire extracellular spike signals, a multi-electrode microdrive assembly consisting of three independently adjustable wire tetrodes was implanted unilaterally above the dorsal striatum, under isoflurane anesthesia[65,66]. Tetrodes consisted of four nichrome wires (12.5 μm, California Fine Wire), which were gold-plated (impedance 0.5–1 MΩ at 1 kHz). After opening small skull holes and reflecting the dura, tetrodes were stereotaxically inserted into the CPu (at bregma, 2.5 mm lateral to midline). Tetrode-implanted mice were tethered to a 16-channel commutator (Plexon) and allowed to move freely in an electrically shielded cage with food and water available ad libitum. Neuronal activity was obtained from the CPu, ranging from 1.5–3.5 mm beneath the dura. A reference electrode was chosen from the electrodes which did not show neuronal activity. Signals from each electrode were band-pass filtered (0.15–8 kHz) and digitized at 40 kHz (MAP system, Plexon) whenever the amplitude of the signal exceeded a preset voltage threshold (usually > 4 SD of peak height distribution). Stainless steel screw electrodes were implanted as above for ECoG/EMG (sampled at 1 kHz). Tetrodes were slowly lowered incrementally to obtain stable recordings. To obtain reliable spike clustering and estimation of neuronal correlates to SWDs, recording time was set typically over 100 min to collect a sufficient number of spikes (usually at least > 1000 spikes). Once the recording session was completed, the electrodes were further advanced over 160 μm to avoid overlap of sampled neurons. Upon completion of the experiment, the electrode tip position was marked by electrolytic lesion (50 μA for 10 s positive to the electrode, negative to ground) and verified histologically using hematoxylin and eosin staining.

**Spike sorting/cell identification.** Single units were isolated offline using multi-dimensional cluster cutting software (Offline sorter, Plexon). Clusters of spikes were attributed to a single unit based on waveform characteristics (peak-valley spike amplitude, 8 principal components of the waveform), excluding contaminated artifacts by checking spike waveforms. Clusters containing spikes with short interspike intervals (<2 ms) exceeding 0.1% were discarded. Only well-isolated clusters with an ellipsoidal shape in multi-dimensional feature space, typical averaged waveform, and a clear refractory period (2 ms) were included in the analysis. Single units stably recorded throughout several sleep/wake cycles were used. Putative MSNs and FSIs were discriminated based on waveform features (spike width and shape)[45] and firing rate (greater than 6 Hz)[44]. Artifactual clusters resulting from double triggering triphasic waveforms were scrutinized and then removed. Cluster drift in time was also checked. After spike sorting, analyses of neuronal activity (raster plot and peri-event time histograms) were performed using NeuroExplorer.

**Statistics.** The data are presented as mean ± SEM. Statistical analyses (two-sided) were performed using Prism 5 (GraphPad Software, La Jolla, CA, USA).

Comparisons between two genotype groups were performed using a Student's *t* test (unpaired), unless otherwise described. When the variance of the data set was significantly different, we used a nonparametric statistical analysis, Mann–Whitney *U* test followed by *t*-test with Welch's correction. *P* values smaller than 0.05 were considered significant.

**Reporting summary**. Further information on research design is available in the Nature Research Reporting Summary linked to this article.

## Data availability

A reporting summary for this Article is available as a Supplementary Information file. The source data underlying Figs. 1a–c, 2a, b, 3a–e, 4b,c, e, 5c, d, 6c, f, 7c, d and 8b, c and Supplementary Figs 1c, 1b, 4, 7, 11a,b,d,e, 12b,c, 14c and 15d-g are provided as a Source Data file. The data in this study are available from the corresponding author upon reasonable request.

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

## Acknowledgements

We are grateful to all members of the Neurogenetics Lab at the RIKEN Center for Brain Science (RIKEN CBS) for helpful discussions, Dr. Yokoyama (RIKEN CBS) for critical reading and comments, and the units for Bio-Material Analysis and Animal Resources Development in the RIKEN CBS Research Resources Division, for neurochemical analyses. This study was supported by the RIKEN CBS, MEXT Grants-in-Aid for Scientific Research (A) (17H01564), AMED under Grant Number JP18dm0107092 (K.Y.); JSPS Grant-in-aid for Young Scientists (B) (21791020) and MEXT Grants-in-Aid for Scientific Research (C) (25461572) (I.O.); MEXT Grants-in-Aid for Scientific Research (C) (15K09848), the Kawano Masanori Memorial Foundation for Promotion of Pediatrics, the Japan Epilepsy Research Foundation, and the JST PRESTO program (H.M.).

## Author contributions

H.M. and K.Y. designed the project. H.M and T.T performed in vivo electrophysiology experiments. T.T. performed acute slice experiments. H.M., A.S., T.Y., T.S., K.A., E.M., M.R., A.O.-A., and T.K.H., performed genetic, histological, biochemical, and pharmacological analyses. I.O., S.I., and K.S. generated mouse lines. K.K, K.K, T.Y., and T.S. prepared virus vectors. H.M. and K.Y. wrote the paper.

## Additional information

**Competing interests:** The authors declare no competing interests.

