## [Peer Review File · Nature Communications]

Reviewers' Comments:

Reviewer #1:

Remarks to the Author:

The manuscript from the Yamakawa lab explores the role of SWDs (spike and wave discharges) on neural circuitry, with a focus on several genetic mutation mouse models and epileptic phenotypes. The authors generated multiple conditional knockout mice and performed a considerable amount of work covering behavior, pharmacological manipulations, and both in vivo and in vitro physiology. While thematically interesting, I could not follow the breadcrumbs of the story. There are many red herrings in the text and I was never quite clear what the question was, even when stated: "we questioned the current paradigm by analyzing circuits mechanisms of SWDs caused by *Stxbp1* and *Scn2a* the risk genes harboring recurrent de novo loss of-function mutations in neurodevelopmental disorders." (Pg 3, line 21). Similarly, the text and figures don't quite match up, such that I couldn't see what the authors stated was there. I have included a few examples of these disconnects below. Of what I could extract from the manuscript, I encourage the authors hone their writing and their points to clear answers to specific questions. It got much better as the paper went on, and keying in on FSI's could be a big finding. I look forward to a re-worked study or studies, but cannot recommend the current manuscript for publication.

Figure 2 – The title states that striatum is a critical node for epilepsy. The results here seem are supposed to show results from 3 different types of inactivations (ssc, cpu, or thalamus). I only see one set of recordings on the left and I cannot tell what to attribute these to - and it is not the local recording of each, as ssc and thalamus are not listed).

Pg 4, line 15 – "assumed to be absence seizures". It is unclear to me what this very important assumption is based upon. The sentence following it might be an attempt to connect it to REM sleep in some fashion, but I don't see a connection.

There is a bounty of facts relayed in this manuscript, but they lack context. The paragraph starting Pg 5, line 14 contains many such facts. I have no idea what connection they have to the study.

pg 3, Line 6 – "suggesting a common pathological mechanism". I appreciate what the authors are attempting, though I think the claim needs to be toned down to what logic can permit. Although ridiculous to say, with this logic, having lungs is a common pathological mechanism.

Figure 7. Average firing rates for FSI's are typically double this rate (see Berke 2011 or several other papers). Additionally, the implication is that roughly 1/4 of recorded neurons in CPU were FSI. This rate is approaching 10x what is typically found.

Reviewer #2:

Remarks to the Author:

This is a very exciting and convincingly documented study that identifies a critical role for decreased corticostriatal excitatory transmission in the generation of non-convulsive epilepsies that are characterized by spike-wave-discharges (SWDs). This is a novel and important insight in the field that has long remained with traditional views provided by rat inbred models of spike-and-wave discharges. In these original studies, a focal area of somatosensory cortex was found to typically act as an initiation site for SWDs and to cause hypersynchronous oscillatory activity in reciprocally connected thalamic circuits, generating the SWDs.

The outstanding insight provided by the present work is that corticostriatal excitatory projections onto

striatal interneurons is causally involved in these SWDs. The work goes beyond the results of a recently published study on the Scn2a-haploinsufficient mice by senior author J Noebels, a world leader in the field of mouse models of absence epilepsy (Commun Biol. 2018;1. pii: 96. doi: 10.1038/s42003-018-0099-2. Epub 2018 Jul 19.).

The paper also excels in the use of a large diversity of techniques that illuminate the impact of corticostriatal transmission deficiency from many critical points of view.

Here are my questions and suggestions:

- 1) The authors should come up with a clearer explanation of how they think abnormal corticostriatal activity ultimately leads to full-blown SWDs that also involve primary thalamocortical circuits. Are the corticostriatal-thalamic loop and the cortico-thalamocortical loops recruited sequentially or in parallel? Is there a time delay between onset of hypersynchrony in the former compared to the latter?
- 2) The lack of effects of Scn2a haploinsufficiency on corticostriatal transmission observed in this study is now becoming clarified through the observation that action potentials are broader in these mice (see the above-mentioned paper from the Noebels group). This could lead to excessive glutamate release followed by a depletion upon repetitive activity. The author could add this as a possibility in the discussion.
- 2) It would be good to mention both mouse lines that were used in the study in the abstract
- 3) To further validate the newly proposed mechanism, it would be interesting to test whether SWDs can be suppressed in a well-established rat line of SWDs such as the GAERS or the WAG/Rij

Reviewer #1

(Reviewer #1 comments)

The manuscript from the Yamakawa lab explores the role of SWDs (spike and wave discharges) on neural circuitry, with a focus on several genetic mutation mouse models and epileptic phenotypes. The authors generated multiple conditional knockout mice and performed a considerable amount of work covering behavior, pharmacological manipulations, and both in vivo and in vitro physiology.

Our response 1

We are grateful to the reviewer for these helpful comments. We thoroughly revised our manuscript, with the changes highlighted in yellow.

(Reviewer #1 comments)

While thematically interesting, I could not follow the breadcrumbs of the story. There are many red herrings in the text and I was never quite clear what the question was, even when stated: “we questioned the current paradigm by analyzing circuits mechanisms of SWDs caused by *Stxbp1* and *Scn2a* the risk genes harboring recurrent de novo loss of-function mutations in neurodevelopmental disorders.” (Pg 3, line 21).

Our response 2

We revised the manuscript at page 3, line 18 as follows:

“There are several rodent models of absence epilepsy in which thalamocortical circuits have been widely accepted as the primary generator of SWDs (Depaulis 2018; Maheshwari 2014). To date, debate continues concerning the critical contributions of cortico-thalamic or thalamo-cortical neurons in generating SWDs within thalamocortical circuits (Depaulis 2018; Sorokin 2017; McCafferty 2018; Bomben 2016). The somatosensory cortex (SSC) has been proposed as the site of initial appearance of SWDs in rat models of absence epilepsy (Meeren 2002; Polack 2007); however, the initial changes responsible for SWD generation and their associated mechanisms have not been fully elucidated.

*In this study, we investigated the initial triggers for SWDs in *Stxbp1*^{+/-} and *Scn2a*^{+/-} mice and their associated circuit mechanisms. Contrary to the previous proposal of the basal ganglia as merely a modulator of SWDs primarily produced by thalamocortical circuits (Deransart 1996; Deransart 1999; Paz, 2007), we here show that impairments in cortico-striatal, rather than cortico-thalamic, pathways trigger SWDs in *Stxbp1*^{+/-} and *Scn2a*^{+/-} mice using multiple experimental approaches.”*

(Reviewer #1 comments)

Similarly, the text and figures don't quite match up, such that I couldn't see what the authors stated was there. I have included a few examples of these disconnects below. Of what I could extract from the manuscript, I encourage the authors hone their writing and their points to clear answers to specific questions. It got much better as the paper went on, and keying in on FSI's could be a big finding. I look forward to a re-worked study or studies, but cannot recommend the current manuscript for publication.

Figure 2 – The title states that striatum is a critical node for epilepsy. The results here seem are supposed to show results from 3 different types of inactivations (ssc, cpu, or thalamus). I only see one set of recordings on the left and I cannot tell what to attribute these to - and it is not the local recording of each, as ssc and thalamus are not listed).

Our response 3

As required, we revised all part of the manuscript. In response to the reviewer's comment, we changed the title of Fig. 2 to "Inactivation of CPU, SSC, and thalamus suppresses SWDs and activation of CPU induces SWDs in *Stxbp1*^{+/-} mice." The left panel shows SWDs in somatosensory cortex (SSC) recordings from mice receiving muscimol injections in five different brain regions, and the right panel shows recordings from the SSC, medial prefrontal cortex (mPFC), and caudate putamen (CPu), where SWDs were strongly observed in *Stxbp1*^{+/-} mice. To clarify these, we revised the manuscript, as follows:

At page 6, line 3;

"We investigated the neural circuits required to generate SWDs in the mutant mice. Local injection of muscimol, a GABAA receptor agonist, into the SSC, CPu, or thalamus but not into the mPFC or hippocampal CA1 region suppressed SWDs in SSC ECoG recordings in *Stxbp1*^{+/-} mice (Fig. 2a, left). In *Stxbp1*^{+/-} mice receiving CPu injection, SWDs were well suppressed not only in the SSC but also in the mPFC and CPu (Fig. 2a, right) where strong SWDs were observed before injection (Fig. 1c). These results demonstrate that neural activity in the SSC, CPu, and thalamus are required for the generation or maintenance of SWDs in *Stxbp1*^{+/-} mice. Although the SSC and thalamus have been well recognized as critical nodes for SWD generation, these results indicate that the CPu is also crucial; thereafter it became a focus of our subsequent experiments. In contrast to muscimol, microinjection of bicuculline, a GABAA receptor antagonist, into the CPu of *Stxbp1*^{+/-} mice induced myoclonic and subsequent generalized convulsive seizures (Supplementary Fig. 5). These data suggest that the CPu is involved in the generation of both absence-like (non-convulsive) and convulsive seizures."

(Reviewer #1 comments)

Pg 4, line 15 – “assumed to be absence seizures”. It is unclear to me what this very important assumption is based upon.

Our response 4

In response to the reviewer’s comment, we eliminated the phrase “...assumed to be absence seizures”, and revised this area at page 4, line 20, as follows:

“Similar to other rodent models of absence epilepsy (Maheshwari 2014; Depaulis 2018) and Scn2a^{+/-} mice (Ogiwara 2018), Stxbp1 knockout mice showed synchronous bilateral cortical SWDs during behavioral quiescence (Fig. 1a) and effective suppression of SWDs following ethosuximide administration (Fig. 1b), therefore they were regarded as experiencing absence seizures.”

(Reviewer #1 comments)

The sentence following it might be an attempt to connect it to REM sleep in some fashion, but I don’t see a connection.

Our response 5

In response to the reviewer’s comment, we revised the manuscript at page 5, line 19, as follows:

“We observed SWDs not only during quiet waking but also during non-rapid eye movement (REM) and REM sleep in Stxbp1^{+/-} mice (Supplementary Fig. 4).”

(Reviewer #1 comments)

There is a bounty of facts relayed in this manuscript, but they lack context. The paragraph starting Pg 5, line 14 contains many such facts. I have no idea what connection they have to the study.

Our response 6

To clarify the context, we added a new figure (Supplementary Fig. 3) and revised the manuscript at page 7, line 12, as follows:

“In contrast, we observed that Stxbp1^{flax/+}/Vgat-Cre (Stxbp1^{fl/+}/Vgat) mice showed twitches (~4 times over 6 hours) and jumps (3–5 times over 6 hours) (Supplementary Video 3) coinciding with ECoG-positive deflections (Supplementary Fig. 3b) but not with SWDs (Fig. 3a, right) or any other epileptic phenotypes. Although mice with a conditional haplo-deletion of Stxbp1 in inhibitory neurons using a Gad2-Cre driver (Gad2-Stxbp1^{cre/+}) have been reported to show severe

epileptic phenotypes and a low survival rate (Kovacevic 2018), we observed that Stxbp1^{flx/+}/Vgat-Cre (Stxbp1^{fl/+}/Vgat) mice showed a normal survival rate, normal growth and locomotor ability (Miyamoto, 2017). Our results clearly indicate that Stxbp1-haploinsufficiency in dorsal-telencephalic excitatory neurons is responsible for SWDs during behavioral quiescence, while the same condition in GABAergic neurons is responsible for the twitches/jumps.”

(Reviewer #1 comments)

pg 3, Line 6 – “suggesting a common pathological mechanism”. I appreciate what the authors are attempting, though I think the claim needs to be toned down to what logic can permit. Although ridiculous to say, with this logic, having lungs is a common pathological mechanism.

Our response 7

In response to the reviewer’s comment, we revised the manuscript at page 3, line 5, as follows: “In particular, STXBPI and SCN2A mutations are common in patients with early-infantile epileptic encephalopathy (Ohtahara syndrome), West syndrome and Lennox-Gastaut syndrome (Saitou 2008; Ogiwara 2009; Nakamura 2013; Wolff 2007; Otsuka 2010) suggesting a potentially overlapping pathological mechanism.”

(Reviewer #1 comments)

Figure 7. Average firing rates for FSI's are typically double this rate (see Berke 2011 or several other papers).

Our response 8

We acknowledge that the mean firing rate of putative fast-spiking interneurons (pFSIs) in our study was relatively low (6.8 Hz at baseline; Fig. 7d in the previous manuscript) as compared with other studies, including the report by Berke et al. (2004). This might be partly explained by our spike-sorting methods, which used strict criteria (Methods) to obtain FSI single units without false-positive contaminated spikes from other units. However, because FSIs exhibit high firing rates *in vivo*, as noted by the reviewer, we introduced firing rates of pFSIs as another strict criteria for discriminating FSIs from other cells. We reanalyzed the peri-event (SWD onset) time histograms of pFSIs with firing rates >6, >8, or >10 Hz at baseline according to the previous study (Thorn & Graybiel 2014) and consistently observed that pFSI firing rates declined at the onset of SWDs as compared with baseline firing (newly added Supplementary Fig. 13h). The decrease in pFSI firing rates at the onset of SWD was statistically significant at their higher firing rates (>6 Hz; mean firing rate: 10.5 Hz) (Fig. 7c

and d), whereas changes in cells with lower firing rates (<6 Hz; mean firing rate: 2.3 Hz) were not significant. These results suggest that cells with higher firing rates and assumed to be FSIs specifically decrease firing at the onset of SWDs. We revised the manuscript accordingly:

At the Results section, page 12, line 8;

“The recorded cells were classified into two neuron types, putative FSIs and MSNs (pFSIs and pMSNs, respectively), based on waveform characteristics (Fig. 7a and Supplementary Fig. 13a–d) and firing rates (Thorn 2014).”

We also updated Fig. 7c and d and Supplementary Fig. 13d-f, h accordingly.

(Reviewer #1 comments)

Additionally, the implication is that roughly 1/4 of recorded neurons in CPu were FSI. This rate is approaching 10x what is typically found.

Our response 9

In response to the reviewer’s comment, we revised the manuscript at page 12, line 12, as follows:

“Although FSIs supposedly constitute <5% of total striatal neurons (Koos 1999; Gittis 2011), we obtained ratios of pFSIs to the total number of the recorded neurons in Stxbp1 mice of 15.9%, which was comparable to other studies (18.5% (Berke 2004) or 10% (Lee 2017)). This preferential detection of FSIs is presumably due to their characteristic short spike width and high firing rate.”

Reviewer #2

Reviewer #2 (Remarks to the Author):

This is a very exciting and convincingly documented study that identifies a critical role for decreased corticostriatal excitatory transmission in the generation of non-convulsive epilepsies that are characterized by spike-wave-discharges (SWDs). This is a novel and important insight in the field that has long remained with traditional views provided by rat inbred models of spike-and-wave discharges. In these original studies, a focal area of somatosensory cortex was found to typically act as an initiation site for SWDs and to cause hypersynchronous oscillatory activity in reciprocally connected thalamic circuits, generating the SWDs.

The outstanding insight provided by the present work is that corticostriatal excitatory projections onto striatal interneurons is causally involved in these SWDs. The work goes beyond the results of a recently published study on the Scn2a-haploinsufficient mice by senior author J Noebels, a world leader in the field of mouse models of absence epilepsy (Commun Biol. 2018;1. pii: 96. doi: 10.1038/s42003-018-0099-2. Epub 2018 Jul 19.).

The paper also excels in the use of a large diversity of techniques that illuminate the impact of corticostriatal transmission deficiency from many critical points of view.

Our response 1

We are grateful for the reviewer's encouraging comments and helpful suggestions. Our changes in the revised manuscript are highlighted in yellow.

(Reviewer #2 comments)

Here are my questions and suggestions:

1) The authors should come up with a clearer explanation of how they think abnormal corticostriatal activity ultimately leads to full-blown SWDs that also involve primary thalamocortical circuits. Are the corticostriatal-thalamic loop and the cortico-thalamocortical loops recruited sequentially or in parallel? Is there a time delay between onset of hypersynchrony in the former compared to the latter?

Our response 2

In response to the reviewer's comment, we re-analyzed the data and added a new figure (Supplementary Fig. 2a,b) to show that SWD peaks in the CPU and thalamus were equally delayed to those in SSC ECoG recordings, which was consistent with the previous observation that SWDs first appeared in the SSC in GARES rats (Polack et al., 2007). However, our results might still support sequential recruitment of corticostriatal and corticothalamic loops. We

revised the manuscript accordingly:

At page 5, line 9;

“SWD peaks in the mPFC, CPu and Thal were equally delayed compared to those observed in SSC ECoG recordings (Supplementary Fig. 2a,b)”.

At page 15, line 8;

“Our results including the reproductions of SWDs by Trpc4-Cre- (cortico-striatal neuron-specific) but not Ntsr1-Cre- (thalamocortical neuron-specific) dependent Stxbp1 or Scn2a deletions or by corticostriatal projection neurons-specific deletion of Stxbp1 (NeuRet) and the induction of SWDs by brief electrical stimulation of the CPu in Stxbp1 mice, support the cortico-striatal pathway as the initial site causally responsible for the seizures and subsequent activation of thalamo-cortical circuits at least in Stxbp1^{+/-} and Scn2a^{+/-} mice, although the cortico-striatal-thalamic loop and cortico-thalamic loop are not mutually exclusive and they might influence SWDs cooperatively.”

(Reviewer #2 comments)

2) The lack of effects of Scn2a haploinsufficiency on corticostriatal transmission observed in this study is now becoming clarified through the observation that action potentials are broader in these mice (see the above-mentioned paper from the Noebels group). This could lead to excessive glutamate release followed by a depletion upon repetitive activity. The author could add this as a possibility in the discussion.

Our response 2

We thank the reviewer for raising this point. We added our interpretation in the Discussion section at page 15, line 3, as follows:

“Additionally, we observed that the effects of Scn2a haploinsufficiency on cortico-striatal transmission were minor (Supplementary Fig. 10f,g). Because Scn2a haploinsufficiency results in the broadening of action potentials (Ogiwara 2018), this could lead to excessive glutamate release followed by a depletion upon repetitive activity. Reduced glutamate transmission in the cortico-striatal pathway is also a possible mechanism in Scn2a^{+/-} mice.”

(Reviewer #2 comments)

2) It would be good to mention both mouse lines that were used in the study in the abstract

Our response 3

We mentioned this in the Abstract.

At page 2, line 5;

*“Mice deficient for *Stxbp1* or *Scn2a* in cortico-striatal but not cortico-thalamic neurons reproduced SWDs.”.*

At page 2, line 13;

*“These findings suggest that impaired cortico-striatal excitatory transmission is a plausible mechanism that triggers epilepsy in *Stxbp1* and *Scn2a* haplodeficient mice.”.*

(Reviewer #2 comments)

3) To further validate the newly proposed mechanism, it would be interesting to test whether SWDs can be suppressed in a well-established rat line of SWDs such as the GAERS or the WAG/Rij

Our response 4

In response to the reviewer’s comment, we obtained GAERS rats as models of absence epilepsy and performed experiments (newly added Fig. 8). Consistent with the circuit model obtained from results of *Stxbp1* and *Scn2a* mice, potentiation of glutamatergic transmission in the CPU or selective reduction in glutamate transmission onto striatal FSIs in the GAERS rats significantly suppressed or facilitated SWDs, respectively. We revised the manuscript, as follows:

In the Results section, page 13, line 3:

*“To investigate whether impaired cortico-striatal excitatory transmission is also observed in animal models of typical absence epilepsy, we tested GAERS rats, a well-established rat strain showing robust and spontaneous SWDs (Danover 1998). Occurrence frequency and duration of SWDs in GAERS rats (Fig. 8a) were larger than those in *Stxbp1*^{+/-} mice (Fig. 1a, Supplementary Fig. 1d). Notably, microinjection of CX516 into the CPU of GAERS rats significantly reduced the number of SWDs (Fig. 8b), whereas NASPM microinjections increased the number of SWDs (Fig. 8c). These data might suggest the generality of our hypothesis that impaired excitatory inputs onto striatal FSIs leads to epilepsy (Fig. 9; see Discussion).”*

Reviewers' Comments:

Reviewer #1:

Remarks to the Author:

I thank the authors for their efforts - the manuscript is improved. My main concern still comes from the conclusions about the role of FSIs, a central tenant of the paper and one that is fatally flawed in multiple ways. The issues, quickly summarized with details to follow are:

- 1) FSI suppression is a common technique, and I am unaware of seizures every being reported
- 2) NASPM, their putative technique for FSI suppression is well documented to have effects on non-FSIs and has never been reported to have induced seizures.
- 3) The cortico-FSI-MSN model demonstrates an unawareness or disregard for basic striatal anatomy.

1) Given the conclusions, it seems that ablating FSIs should lead to seizures. This would be an excellent positive control, and in fact it's already been done -Xu et al., 2016 ('Ablation of fast-spiking interneurons in the dorsal striatum, recapitulating abnormalities seen post-mortem in Tourette syndrome, produces anxiety and elevated grooming'). The fact that this very relevant study is not mentioned is a bit concerning, but one cannot read everything.

However, FSI ablation and/or suppression is arguably the most common mouse model for Tourette's. I'm not aware of a single case of seizures in the above study or in any other study using FSI suppression. Given their results, it is surprising that no one has shown this, and I personally know several lab heads who use this FSI suppression model. This is a very serious issue that does not seem to be accounted for either in the experiments nor the discussion.

2) Line 232: "we injected 1-naphthyl acetyl spermine (NASPM), a selective blocker of calcium-permeable AMPA receptors and abundantly expressed in striatal FSIs but not in MSNs, into the CPu of WT mice (Fig. 6a)."

Any citation would be good to back up this rather critical point. NASPM is well-known to be expressed in CHaT neurons of the striatum as well as FSI's. There are also several papers - including one in Nature ("Formation of accumbens GluR2-lacking AMPA receptors mediates incubation of cocaine craving", 2008 - also see McCutcheon 2011) that DO find an effect of NASPM on MSNs! They also report zero seizure incidents. An explanation is necessary.

3) Their model is based upon a striatum in which direct pathway neurons, which comprise almost half of the neurons of the striatum, play no role (see figure 9 and page 14). The motivation for this was stated:

Line 306: "Cortico-striatal inputs predominantly activate enkephalin-positive MSNs in the indirect pathway⁵², suggesting MSN-GPe as a major route for the seizures."

Even though I would say this 22 year-old model is out of date (see Wall et al., Neuron 2013), a bias in cortico-striatal input doesn't matter:

- The model hinges not upon cortico-MSN input, but rather cortico-FSI input. There is no evidence, cited here or in any literature that I am aware of, that FSI's project only onto indirect pathway MSNs - as their model explicitly suggests. Their model, discussion, and the inclusion of the citation suggest a lack of basic understanding of the circuit.
- Direct pathway neurons are heavily and reciprocally connected with indirect pathway neurons. Even if FSI's projected only onto indirect pathway neurons, an account of indirect to direct, and the direct pathway's very strong effect on SNr activity, needs to be taken into account.
- Direct and indirect pathway MSNs are considered together in every instance in the paper. Why are the neurons indirect pathway MSNs now, other than convenience? There are various methods (transgenic (e.g. drd1a, adora2a, or drd2 cre lines) or tracer injections into the different target nuclei)

to identify either population should the authors wish to suggest this mechanism. It seems odd that the effect is supposed to be mediated through MSNs, but in my reading they show no effect.

In short, the model and motivation for the observed effects has several serious flaws and cannot be considered to be a putative mechanism.

More minor comments:

In Supplemental Figure 2 b, with an n of 5, there does not appear to be a normal distribution of mPFC delays thus a t-test should not be used to test the difference between different brain areas.

Where are the data supporting the claims made on lines 163 – 166?

Line 60 – recurrent seems odd. Please change to “common”, or the like

Reviewer #2:

Remarks to the Author:

This exciting paper has been excellently revised. I appreciate in particular the additional data in Suppl Fig 2 regarding the timing of SWDs in CPU and Thal. Additionally, the authors now add data on the GAERS model that confirm their hypothesis. This is a true step forward in the field. I have no further comments.

Anita Luthi

REVIEWERS' COMMENTS:

Reviewer #1 (Remarks to the Author):

I thank the authors for their efforts - the manuscript is improved. My main concern still comes from the conclusions about the role of FSIs, a central tenant of the paper and one that is fatally flawed in multiple ways. The issues, quickly summarized with details to follow are:

- 1) FSI suppression is a common technique, and I am unaware of seizures every being reported
- 2) NASPM, their putative technique for FSI suppression is well documented to have effects on non-FSIs and has never been reported to have induced seizures.
- 3) The cortico-FSI-MSN model demonstrates an unawareness or disregard for basic striatal anatomy.

Our response 1

We thank Reviewer #1 for their important and thoughtful comments, which have improved our manuscript. We have revised the manuscript to respond to all concerns raised by the reviewer as described below.

1) Given the conclusions, it seems that ablating FSIs should lead to seizures. This would be an excellent positive control, and in fact it's already been done -Xu et al., 2016 ('Ablation of fast-spiking interneurons in the dorsal striatum, recapitulating abnormalities seen post-mortem in Tourette syndrome, produces anxiety and elevated grooming'). The fact that this very relevant study is not mentioned is a bit concerning, but one cannot read everything.

However, FSI ablation and/or suppression is arguably the most common mouse model for Tourette's. I'm not aware of a single case of seizures in the above study or in any other study using FSI suppression. Given their results, it is surprising that no one has shown this, and I personally know several lab heads who use this FSI suppression model. This is a very serious issue that does not seem to be accounted for either in the experiments nor the discussion.

Our response 2

In addition to the experiments in Xu et al. (2016), pharmacological inhibition of striatal fast-spiking interneurons (FSIs) (Gittis et al., 2011) and optogenetic inhibition of striatal FSIs (Lee et al., 2017) have been performed in mice. Neither spike-and-wave discharges (SWDs) nor epileptic seizures were described in these reports; however, Klaus and Plenz (2016) found SWD-like cortical local field potential (LFP) changes following pharmacological FSI inhibition, which is similar to our observation. In particular, non-convulsive seizures such as absence seizures are very difficult to detect without electroencephalographic recordings. Even *Stxbp1* mice, *Scn2a* mice, and GAERS rats showing frequent SWDs appear outwardly normal. It is likely that in previous studies not designed to detect epilepsies, SWDs, or other epileptic brain activity, these may have been

overlooked.

We have explained these points in the revised manuscript (Discussion, line 314):

“Pharmacological suppression (Gittis 2011), cell ablation (Xu 2016), and optogenetic suppression of striatal FSIs (Lee 2017) have been performed in mice. Neither SWDs nor epileptic seizures were described in these reports; however, Klaus and Plenz (2016) found SWD-like cortical LFP changes following pharmacological FSI inhibition, which is similar to our observation. In particular, non-convulsive seizures such as absence seizures are very difficult to detect without electroencephalographic recordings. Even *Stxbp1* mice, *Scn2a* mice, and GAERS rats showing frequent SWDs appear outwardly normal. It is likely that in previous studies not designed to detect epilepsies, SWDs, or other epileptic brain activity, these may have been overlooked.”

2) Line 232: “we injected 1-naphthyl acetyl spermine (NASPM), a selective blocker of calcium-permeable AMPA receptors and abundantly expressed in striatal FSIs but not in MSNs, into the CPu of WT mice (Fig. 6a).”

Any citation would be good to back up this rather critical point.

Our response 3

We have cited three references (Deng et al., 2007; Gittis et al., 2011; Klaus and Plenz, 2016) to support this description (line 248).

NASPM is well-known to be expressed in CHaT neurons of the striatum as well as FSIs. There are also several papers – including one in Nature (“Formation of accumbens GluR2-lacking AMPA receptors mediates incubation of cocaine craving”, 2008 – also see McCutcheon 2011) that DO find an effect of NASPM on MSNs! They also report zero seizure incidents. An explanation is necessary.

Our response 4

According to the report by Gittis et al. (2011), in which the effects of a calcium-permeable AMPA receptor blocker on striatum neurons including MSNs were assessed, FSIs and cholinergic neurons were tested. Although excitatory postsynaptic currents were decreased in cholinergic neurons (~45%) and FSIs (~73%) after application of the blocker, the firing rates of cholinergic neurons were not altered in vitro. Rather, the firing rates of FSIs were only selectively decreased after application of the blocker in vivo. We observed that upregulation of FSI activity using the DREADD system rescued the SWD phenotype of *Stxbp1* mice (Fig. 6) and that temporal downregulation of FSI occurred at the onset of

SWDs (Fig. 7), supporting the critical role of FSIs in SWDs. However, we could not exclude the possibility that cholinergic interneurons were affected by NASPM.

Based on this, we have revised our manuscript as follows (line 309):

“Gittis et al. reported that the application of a calcium-permeable AMPA receptor blocker to the striatum decreased excitatory postsynaptic currents in cholinergic neurons (~45%) and FSIs (~73%), although the firing rates of only FSIs, and not cholinergic neurons, were selectively decreased. However, we still cannot exclude the possibility that epileptic activity caused by NASPM was partially mediated by cholinergic interneurons.”

As mentioned by the reviewer, Conrad et al. (2008) observed decreased excitatory post-synaptic current amplitudes in striatal MSNs after NASPM application; however, NASPM only affected cocaine-exposed rats and not saline-exposed control rats. McCutcheon et al. (2011) confirmed the lack of an effect of NASPM on MSNs in control animals. Again, these studies focused on cocaine-seeking behavior, and therefore, SWDs or other epileptic brain activity might not have been recognized.

3) Their model is based upon a striatum in which direct pathway neurons, which comprise almost half of the neurons of the striatum, play no role (see figure 9 and page 14). The motivation for this was stated:

Line 306: “Cortico-striatal inputs predominantly activate enkephalin-positive MSNs in the indirect pathway52, suggesting MSN-GPe as a major route for the seizures.”

Even though I would say this 22 year-old model is out of date (see Wall et al., Neuron 2013), a bias in cortico-striatal input doesn't matter:

- The model hinges not upon cortico-MSN input, but rather cortico-FSI input. There is no evidence, cited here or in any literature that I am aware of, that FSI's project only onto indirect pathway MSNs – as their model explicitly suggests. Their model, discussion, and the inclusion of the citation suggest a lack of basic understanding of the circuit.

- Direct pathway neurons are heavily and reciprocally connected with indirect pathway neurons. Even if FSI's projected only onto indirect pathway neurons, an account of indirect to direct, and the direct pathway's very strong effect on SNr activity, needs to be taken into account.

- Direct and indirect pathway MSNs are considered together in every instance in the paper.

Why are the neurons indirect pathway MSNs now, other than convenience? There are various methods (transgenic (e.g. drd1a, adora2a, or drd2 cre lines) or tracer injections into the different target nuclei) to identify either population should the authors wish to suggest this mechanism. It seems odd that the effect is supposed to be mediated through MSNs, but in my reading they

show no effect.

In short, the model and motivation for the observed effects has several serious flaws and cannot be considered to be a putative mechanism.

Our response 5

We would like to note that we are not proposing that FSIs project only onto indirect pathway MSNs, but we are rather simply suggesting that the MSN-globus pallidus externus (GPe) would be a major route for the seizures, according to our experimental results. Based on these, we propose that dysfunction of FSIs is one of the potential causes of SWDs. Previous studies have also shown that pharmacological manipulation of the GPe or subthalamic nucleus (STN), structures in the indirect pathway, influence SWDs (see the Discussion in our manuscript). The paper cited by the reviewer (Wall et al., Neuron 79, 347–360, 2013) reported that "sensory cortical structures preferentially innervated the direct pathway, whereas motor cortex preferentially targeted the indirect pathway", although the preference was rather moderate, and their analysis was based on histology and not on electrophysiology.

However, we still acknowledge that further studies are needed to validate our indirect pathway model for SWDs. As pointed out by the reviewer, direct/indirect pathway-selective manipulation and a more detailed analysis of neuronal interactions among the cortex, basal ganglia, and thalamus are required in future studies to confirm our model.

We have explained these limitations of our model and proposed future directions for the research in the discussion (line 338):

“However, because there are reciprocal connections between direct and indirect pathway neurons, it also remains to be determined how the disinhibition of FSIs preferentially affects SWDs via the indirect pathway. Direct/indirect pathway-selective manipulation and a more detailed analysis of neuronal interactions among the cortex, basal ganglia, and thalamus are required in future studies to confirm our model”.

More minor comments:

In Supplemental Figure 2 b, with an n of 5, there does not appear to be a normal distribution of mPFC delays thus a t-test should not be used to test the difference between different brain areas.

Our response 6

In response to the reviewer's comment, we used the Mann-Whitney test instead of the t-test and have revised the manuscript accordingly (Supplementary fig. 2 legend).

Where are the data supporting the claims made on lines 163 – 166?

Our response 7

We apologize for citing the wrong reference here. The correct reference supporting the claim is Ogiwara et al., Commun. Biol. (2018). We have revised the manuscript accordingly (line 179).

Line 60 – recurrent seems odd. Please change to “common”, or the like

Our response 8

We have changed “recurrent” to “common” (line 73).

Reviewer #2 (Remarks to the Author):

This exciting paper has been excellently revised. I appreciate in particular the additional data in Suppl Fig 2 regarding the timing of SWDs in CPu and Thal. Additionally, the authors now add data on the GAERS model that confirm their hypothesis. This is a true step forward in the field.

I have no further comments.

Anita Luthi

Our response

We sincerely appreciate your supportive comment regarding our manuscript.